# Boosting reactivity of water-gas shift reaction by synergistic function over $CeO_{2-x}$/$CoO_{1-x}$/Co dual interfacial structures

Xin-Pu Fu[1,4], Cui-Ping Wu[1,4], Wei-Wei Wang [1], Zhao Jin[1], Jin-Cheng Liu[2 ✉], Chao Ma [3 ✉] & Chun-Jiang Jia [1 ✉]

Dual-interfacial structure within catalysts is capable of mitigating the detrimentally completive adsorption during the catalysis process, but its construction strategy and mechanism understanding remain vastly lacking. Here, a highly active dual-interfaces of $CeO_{2-x}$/$CoO_{1-x}$/Co is constructed using the pronounced interfacial interaction from surrounding small $CeO_{2-x}$ islets, which shows high activity in catalyzing the water-gas shift reaction. Kinetic evidence and in-situ characterization results revealed that $CeO_{2-x}$ modulates the oxidized state of Co species and consequently generates the dual active $CeO_{2-x}$/$CoO_{1-x}$/Co interface during the WGS reaction. A synergistic redox mechanism comprised of independent contribution from dual functional interfaces, including $CeO_{2-x}$/$CoO_{1-x}$ and $CoO_{1-x}$/Co, is authenticated by experimental and theoretical results, where the $CeO_{2-x}$/$CoO_{1-x}$ interface alleviates the CO poison effect, and the $CoO_{1-x}$/Co interface promotes the $H_2$ formation. The results may provide guidance for fabricating dual-interfacial structures within catalysts and shed light on the mechanism over multi-component catalyst systems.

Previous findings have validated that the surface reaction normally proceeded at the metal-support interface upon the numerous hybrid catalysts[1–4]. The most well-known strategy is anchoring metal atoms onto the surface of oxide support to establish mono-interfacial structure[5–7], while the ubiquitously competitive adsorption between reactant and product molecules would inevitably interfere with catalytic efficiency[8,9]. From this perspective, a strategy based on the synergistic participation of dual interfaces is validated as an efficient path for catalyst design targeted for industrial catalysis processes[3,10]. However, the efficiently targeted fabrication of the dual interface is impeded by the intricacy of the multifarious sorption ability desired for the catalysis process[11–14].

As a reducible transition metal, cobalt is promising for dual-interface design based on the prerequisite that dual kinds of Co species could integrate the advantages of each component[14–16]. The

metallic $Co^0$ atom is commonly regarded as an active site for activating CO molecules in kinds of catalysis processes owing to its intense CO adsorption energy and electron-donor character[17]. The importance of $Co^{2+}$ to the optimum adsorption/coverage of reactants or intermediates has been gradually recognized in recent studies[14,15,18]. The co-presence of Co and $CoO_x$ on the surface of $Al_2O_3$ or $ZrO_2$ was favorable for $CO/CO_2$ activation and C-H bond scission[19,20]. The structural engineering of Co-based catalysts aiming to establish highly active and stable dual-interface targeting to specific catalysis processes is thus desirable. However, fabricating a stable abundance $Co/CoO_x$ interface within an analogous $Co/CoO_x/MO_x$ structure is severely impeded by the unstable feature of $Co^{2+}$(CoO) species, especially under a reductive atmosphere[11,12,21]. Meanwhile, the respective role of each interface and the corresponding synergistic mechanism have been insufficiently substantiated so far[12–15,22]. A possible approach to overcoming the

[1]Key Laboratory for Colloid and Interface Chemistry, Key Laboratory of Special Aggregated Materials, School of Chemistry and Chemical Engineering, Shandong University, 250100 Jinan, China. [2]Center for Rare Earth and Inorganic Functional Materials, School of Materials Science and Engineering & National Institute for Advanced Materials, Nankai University, 300350 Tianjin, China. [3]College of Materials Science and Engineering, Hunan University, 410082 Changsha, China. [4]These authors contributed equally: Xin-Pu Fu, Cui-Ping Wu. ✉e-mail: liujincheng@nankai.edu.cn; cma@hnu.edu.cn; jiacj@sdu.edu.cn

above obstacles is modulating the oxidized state of metal atoms via modifying the metal-support interaction[23,24].

CeO$_2$ is encouraging for enhancing the stability of CoO$_x$ species and establishing efficient dual interfacial Co-based catalysts because the defective structures (O$_{v(CeO2)}$) derived from spontaneous Ce$^{3+}$/Ce$^{4+}$ change, which are prone to provide activated O species and thus more readily modulate the oxidized state of atoms at the interface[25]. In this work, we reported the crucial role of CeO$_{2-x}$ in maintaining the CoO phase under reductive conditions, resulting in the fabrication of a stable CeO$_{2-x}$/CoO$_{1-x}$/Co dual-interfaces structure. The as-formed CeO$_{2-x}$/CoO$_{1-x}$/Co structure efficiently catalyzes the water gas shift (WGS) reaction (CO + H$_2$O ⇌ CO$_2$ + H$_2$), a vital process for both model catalysis and hydrogen upgrading applications[13,14,26–28]. The in situ characterization and DFT calculation disclosed that CeO$_{2-x}$/CoO$_{1-x}$ and Co/CoO$_{1-x}$ interfaces are synergistically involved in reaction cycles. The findings in this work provide a strategy to optimize the sorption process upon the catalyst surface via dual-interfacial engineering.

## Result and discussion
### Catalytic performance of CeCoO$_x$ catalysts
The CeCoO$_x$ catalysts were prepared through an ultrasonic spray approach (Supplementary Fig. 1) with tunable atomic ratios of Ce/Co for the raw materials (Ce/Co = 1/9, 9/1 and 0/10), labeled by 1Ce9CoO$_x$, 9Ce1CoO$_x$ and Co$_3$O$_4$ respectively[29,30]. The overall structure illustrated in transmission/scanning electron microscopy (TEM/SEM) images (Fig. 1a and Supplementary Fig. 2) revealed that the Co and Ce species were assembled into a spherical framework. As shown in Supplementary Fig. 3, apart from the weak diffracted peaks for the Co$_3$O$_4$ phase, non-detectable crystalized CeO$_2$ can be identified in the XRD patterns of the fresh 1Ce9CoO$_x$ catalyst, suggesting the small crystalized size of CeO$_2$. The elemental mapping results (Supplementary Fig. 4) further illustrated the Co and Ce species homogenously distributed over the 1Ce9CoO$_x$ sample. In contrast with pristine Co$_3$O$_4$ catalyst, the averaged Co$_3$O$_4$ crystal size in the 1Ce9CoO$_x$ determined by the Scheler equation is much smaller (ca. 15.4 vs. 6.1 nm), indicating the influential role of the minor CeO$_2$ species in stabilizing and dispersing predominantly present Co species.

The as-prepared samples were evaluated as catalysts for the WGS reaction after pretreating by 5%H$_2$/Ar at 400 °C for 1 h. The CO conversions were valued with the WGS reaction temperatures elevated from 180 to 320 °C. In detail, the CO conversion of 1Ce9CoO$_x$ reached 28% at ca. 240 °C with a high gas hourly space velocity (GHSV) of 168,000 mL g$^{-1}$ h$^{-1}$ (Supplementary Fig. 5), which was tremendously better than that of the Co$_3$O$_4$ catalyst (3%). This gap in CO conversion between the 9Ce1CoO$_x$ and bare Co$_3$O$_4$ catalysts was further widened with a relatively low GHSV of 42,000 mL g$^{-1}$ h$^{-1}$ at ca. 240 °C (95% $vs.$ 14% as illustrated in Fig. 1b). Otherwise, the pristine CeO$_2$ was almost inactive in catalyzing the WGS reaction as the reaction temperature below 300 °C, suggesting that the momentous synergetic effect present between CeO$_2$ and Co-related species. This hypothesis is further evidenced by the kinetic experiments. The apparent activation energy ($E_a$) value was distinctly decreased (Supplementary Fig. 6) for the 1Ce9CoO$_x$ catalyst (100.7 kJ/mol) as compared with that of the Co$_3$O$_4$ catalyst (188.4 kJ/mol), implying that the reactant molecules might react through an advantageous reaction pathway upon the 1Ce9CoO$_x$ catalyst.

The 1Ce9CoO$_x$ catalyst demonstrated a remarkable mass-normalized reaction rate of 169.3 mmol$_{CO}$/g$_{cat}$/h at 250 °C (Fig. 1c, Supplementary Table 1), 3-folds of the value of 50.1 mmol$_{CO}$/g$_{cat}$/h for the 1Al9CoO$_x$ catalyst with irreducible Al$_2$O$_3$ species, indicative of the exclusive role of reducible CeO$_2$ in activating Co species. The boosted activity of 1Ce9CoO$_x$ catalyst was also higher than commercial CuZnAl catalyst and noble Au-Cluster/CeO$_2$ catalyst (64.8 and 27.8 mmol$_{CO}$/g$_{cat}$/h, Supplementary Table 1) under the same reaction condition[31,32]. The 1Ce9CoO$_x$ catalyst revealed good stability during the long-term

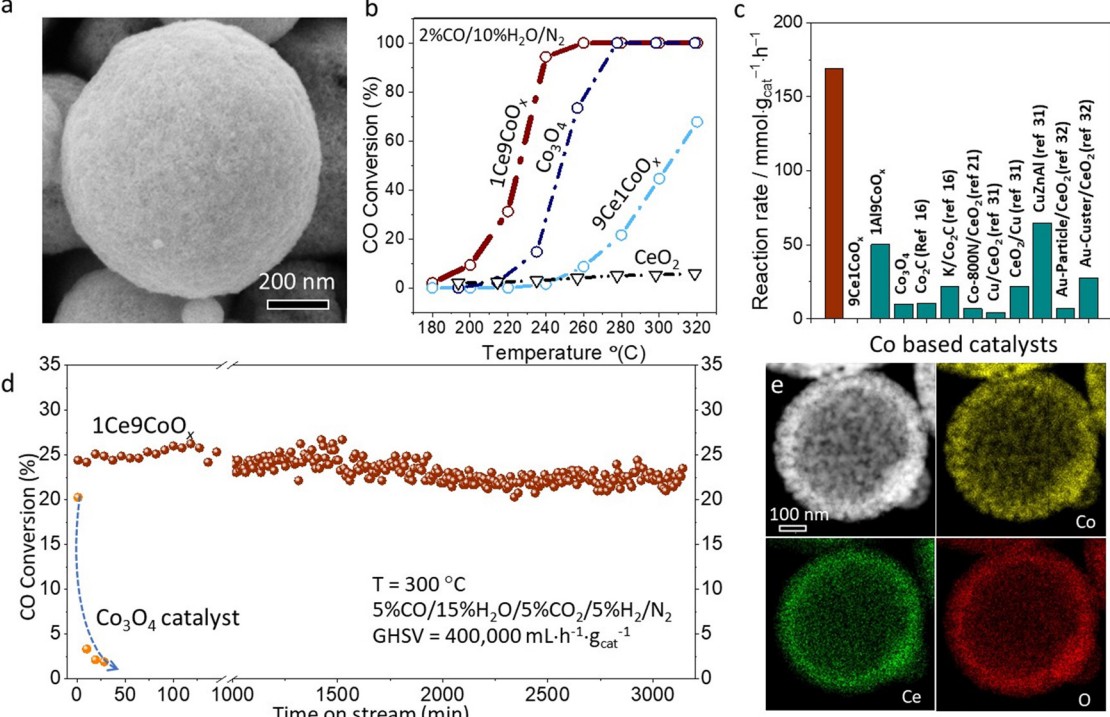

**Fig. 1 | The catalytic performance and structural characterization of catalysts. a** SEM image of fresh 1Ce9CoO$_x$ catalyst. **b** CO conversions of WGS reaction over the 1Ce9CoO$_x$, Co$_3$O$_4$ and 9Ce1CoO$_x$ catalysts at various temperatures. Reaction condition: 2%CO/10%H$_2$O/N$_2$, GHSV = 42,000 mL/g$_{cat}$/h. **c** Comparison of the reaction rate with reference catalysts at 250 °C[16,21,31,32]. The detailed reaction conditions and CO conversions for these catalysts are summarized in Supplementary Table 1. **d** Stability test results of the 1Ce9CoO$_x$ and Co$_3$O$_4$ catalysts evaluated at 300 °C. Other reaction condition: 5%CO/15%H$_2$O/5%H$_2$/5%CO$_2$/N$_2$, GHSV = 400,000 mL g$_{cat}^{-1}$ h$^{-1}$. **e**, HAADF-STEM image and X-ray EDS elemental mappings (Co, Ce, and O) for the spent 1Ce9CoO$_x$ catalyst.

test at 250 °C for 1800 min with a mild reaction atmosphere of (2%CO/ 10%$H_2O$/$N_2$), which outperformed the performance of the $Co_3O_4$ catalyst (Supplementary Fig. 7). To be noted, the gap in stability between 1Ce9CoO$_x$ and $Co_3O_4$ catalyst was more significant under a relatively harsh reaction condition ($T$ = 300 °C, 5%CO/15%$H_2O$/5%$H_2$/5%$CO_2$/$N_2$, GHSV = 400,000 mL g$_{cat}^{-1}$ h$^{-1}$), where the 1Ce9CoO$_x$ was relatively stable over 3000 min on stream and the bare $Co_3O_4$ catalyst rapidly deactivated from 20% to 2% in the initial 40 min (Fig. 1d). The spent 1Ce9CoO$_x$ catalysts were primarily explored by the aberration-corrected high-angle annular dark-field scanning transmission electron microscopy (ac-HAADF STEM). Supplementary Fig. 8 depicted that the nanoparticles within the spent 1Ce9CoO$_x$ catalyst maintained assembled sphere morphology. Elemental mapping results proved that the Co and Ce species distributed homogeneously after either transient (Fig. 1e and Supplementary Fig. 9) or a long-term WGS reaction test (Supplementary Fig. 10). The average diameter of $CeO_2$ and CoO species derived from statistical results based on the HRTEM images were about 4.0 and 4.3 nm, where the lattice distance of all nanoparticles was verified prior to counting (Supplementary Figs. 11, 12). The aforementioned results evidenced the essential role of introduced $CeO_2$ species in enhancing the catalytic efficiency and stability during catalyzing WGS reaction for the CeCoO$_x$ catalyst.

## Identification of the CeO$_{2-x}$/CoO$_{1-x}$/Co dual-interface structure

The as-formed CeO$_{2-x}$ nanoparticles with a ca. 2–4 nm diameter were located neighboring Co species (Fig. 2a–c, Supplementary Fig. 13), which were in line with the mean size as derived from HRTEM images or the XRD results based on the Scherrer Formula ($CeO_2$: ~4.3 nm and CoO: ~4.2 nm). In sharp comparison, the nanoparticles within the $Co_3O_4$ sample seriously aggregated after the long-term WGS reaction, as shown in Supplementary Fig. 14, suggesting that the presence of $CeO_2$ species effectively inhibits the excessive crystallization of Co species under the WGS reaction condition. In most cases, the optimal metal-oxides interface was commonly achieved by anchoring highly dispersed metal sites onto the surface of oxide support. In contrast, the inevitable growth of metal species, especially under reductive conditions, results in the irreversible loss of specific sites at the interface[8,9]. The amount of the $CeO_2$-Co interfacial sites within the 1Ce9CoO$_x$ catalyst was defined by the perimeter outline of small CeO$_{2-x}$ particles with favored thermostability, which consequently resist the loss of interfacial Co-CeO$_2$ sites under reaction conditions.

The coexistence of metallic Co and CoO was also demonstrated by the XRD and XAFS results as shown in Supplementary Fig. 15 and Fig. 16. Lattice distances of 0.20, 0.24, and 0.27 nm respectively ascribed to Co(111), CoO(111) and $CeO_2$(200) were identified as shown in Fig. 2a, where the dominant CoO phase was identified in the region between metallic Co and $CeO_2$ (Fig. 2a, b) or directly anchored with $CeO_2$ nanoparticles (Supplementary Fig. 13). Similarly, as proven by the elemental mapping results, the differentiated distribution of Ce and Co implied that the $CeO_2$ islets interspersed among the Co species (Fig. 2c). The XPS spectra collected before and after light-off WGS reaction was collected and deconvoluted. As shown in Fig. 2d, the fresh 1Ce9CoO$_x$ catalyst was characterized by the Co 2p$_{3/2}$ binding energy at 779.9 eV and a shake-up satellite signal at 790.4 eV with low intensity, which is the typical feature for the Co$^{2+}$/Co$^{3+}$ ions in the $Co_3O_4$ spinel structure. The primary presence of the $Co_3O_4$ phase for the fresh 1Ce9CoO$_x$ catalyst was also confirmed by the Raman results, as shown in Supplementary Fig. 3b. After the light-off WGS reaction (2%CO/10% $H_2O$/Ar), the binding energy of Co species was observed at 780.1 eV coupling with a strong satellite peak at 6.1 eV higher, which was typically ascribed to CoO phase[33]. Only Co$^{2+}$ ions in octahedral sites of rock-salt CoO may result in these typical photoemission features, where the Co$^{2+}$ ions in $Co_3O_4$ do not locate on octahedral sites[14]. The spin-orbit coupling peak at ca. 16.0 eV higher was also a characteristic feature for CoO[33], which could be clearly identified for the spent

1Ce9CoO$_x$ catalyst. In addition, a small shoulder peak at 778.0 eV ascribed to metallic Co was detected after the WGS reaction. Based on the above comparison, an unambiguous change from $Co_3O_4$ to Co/ CoO upon the surface of the 1Ce9CoO$_x$ catalyst was induced by the pretreatment and the WGS reaction. The Ce 3d spectrum was deconvoluted and labeled according to Burroughs formalism (Fig. 2e), where the fitted u′ and u$^0$ resulted from Ce$^{3+}$. Interestingly, compared with fresh catalyst, the Ce$^{3+}$ fraction was significantly boosted from 0.15 to 0.28, indicating that the amount of O$_v$-Ce increased after the WGS reaction. A similar conclusion could be derived from the O 1s results (Fig. 2f), in which the vacancy-related O species (O2 and O3) were increased from 0.40 to 0.68 after the WGS reaction[34]. In addition, it should be noted that the relative amount of Ce$^{3+}$ species upon the surface layer as determined by XPS results was lower than 0.03 (Ce$^{3+}$/ (CeO$_x$ + CoO$_x$)), suggesting that the O species related with O$_v$-Ce$^{3+}$ structure should be a minor component. Videlicet, this discrepancy indicated the plausible presence of CoO$_{1-x}$ species upon the surface of the 1Ce9CoO$_x$ catalyst.

To exclude the plausible interference from oxidation, quasi in situ XPS experiments were conducted over the 1Ce9CoO$_x$ catalyst (2%CO/3% $H_2O$/Ar, 250 °C) to further estimate the reconstruction process of Co species during the WGS reaction. Supplementary Fig. 17a shows the Co 2p photoemission feature of 1Ce9CoO$_x$ catalyst where the robust satellite peaks at ca. 786.2 eV indicate that the dominant presence of CoO on the surface layer of 1Ce9CoO$_x$ was generated during the WGS reaction instead of being oxidized by the air. The Ce$^{3+}$ content (μ$^0$ + μ′) as a percentage of total cerium content was calculated to be 0.24 over 1Ce9CoO$_x$ catalyst after in situ WGS reaction (Supplementary Fig. 17b), proving the abundant generation of O$_{v(CeO2)}$ under reaction condition[35]. The corresponding CO conversions for various catalysts were determined with the comparable conditions (260 °C, 2%CO/3%$H_2O$/Ar), where the 1Ce9CoO$_x$ catalyst demonstrated much better catalytic performance than the reference catalysts (Supplementary Fig. 18).

The presence of defective sites over the 1Ce9CoO$_x$ and bare $Co_3O_4$ catalysts was further explored by the Raman experiments. The peaks at 191, 484, and 690 cm$^{-1}$ were detectable for fresh 1Ce9CoO$_x$ catalyst (Supplementary Fig. 3), which were attributed to $F_{2g}$, $E_g$ and $A_{1g}$ symmetry modes of the crystalline $Co_3O_4$ respectively[36]. After in situ feeding 2%CO/3%$H_2O$/Ar upon the surface of 1Ce9CoO$_x$ catalyst at 250 °C, three peaks at regions of ca. 450, 530–550, and 580 cm$^{-1}$ were observed as shown in Fig. 2g, whereas they were absent for pristine $Co_3O_4$ catalyst (Supplementary Fig. 19). According to previous reports, the signals at ca. 450 and 580 cm$^{-1}$ could be ascribed to the $F_{2g}$ mode and defect-induced mode (D1) of the ceria fluorite phase[37–40]. There are two plausible affiliations for the signal at the region of 530–550 cm$^{-1}$: it might be derived from the oxygen vacancies within the surface or subsurface layer of $CeO_2$ nanoparticles (surface O$_{v(CeO2)}$)[37]; or the possible presence of oxygen vacancy in CoO phase (O$_{v(CoO)}$) contribute to this Raman signal[36]. On the basis of the predominant presence of Co$^{2+}$ in CoO identified by XPS results (Fig. 2d), we tend to attribute the Raman peak at the region of 530–550 cm$^{-1}$ to the combinative signals of O$_{v(CeO2)}$ and O$_{v(CoO)}$ within the CeO$_{2-x}$/CoO$_{1-x}$ interface region. As shown in Supplementary Fig. 20, the presence of the O$_v$(CoO) was also supported by the EELS results for the spent 1Ce9CoO$_x$ catalyst. The calculated $L_3$/$L_2$ ratio for Co $L$-edge collected in the Co-rich region was ca. 3.2. In contrast, the corresponding values calculated for regions overlapped with $CeO_2$ increased to ca. 4.8. The intensity ratio of $L_3$/$L_2$ is determined by the occupation state of 3d-states, which the increased $L_3$/$L_2$ value might thus indicate the lower valence state of Co species induced by the creation of oxygen vacancies within the interface region[34]. The facile formation of oxygen vacancies on CoO$_{1-x}$ could result from a larger Co−O bond length and relatively weak bond strength in rock-salt CoO, in contrast to those of $Co_3O_4$[14]. In addition, the presence of CeO$_{2-x}$/CoO$_{1-x}$ and Co/CoO$_{1-x}$ interface within the spent 1Ce9CoO$_x$ catalysts was carefully substantiated by

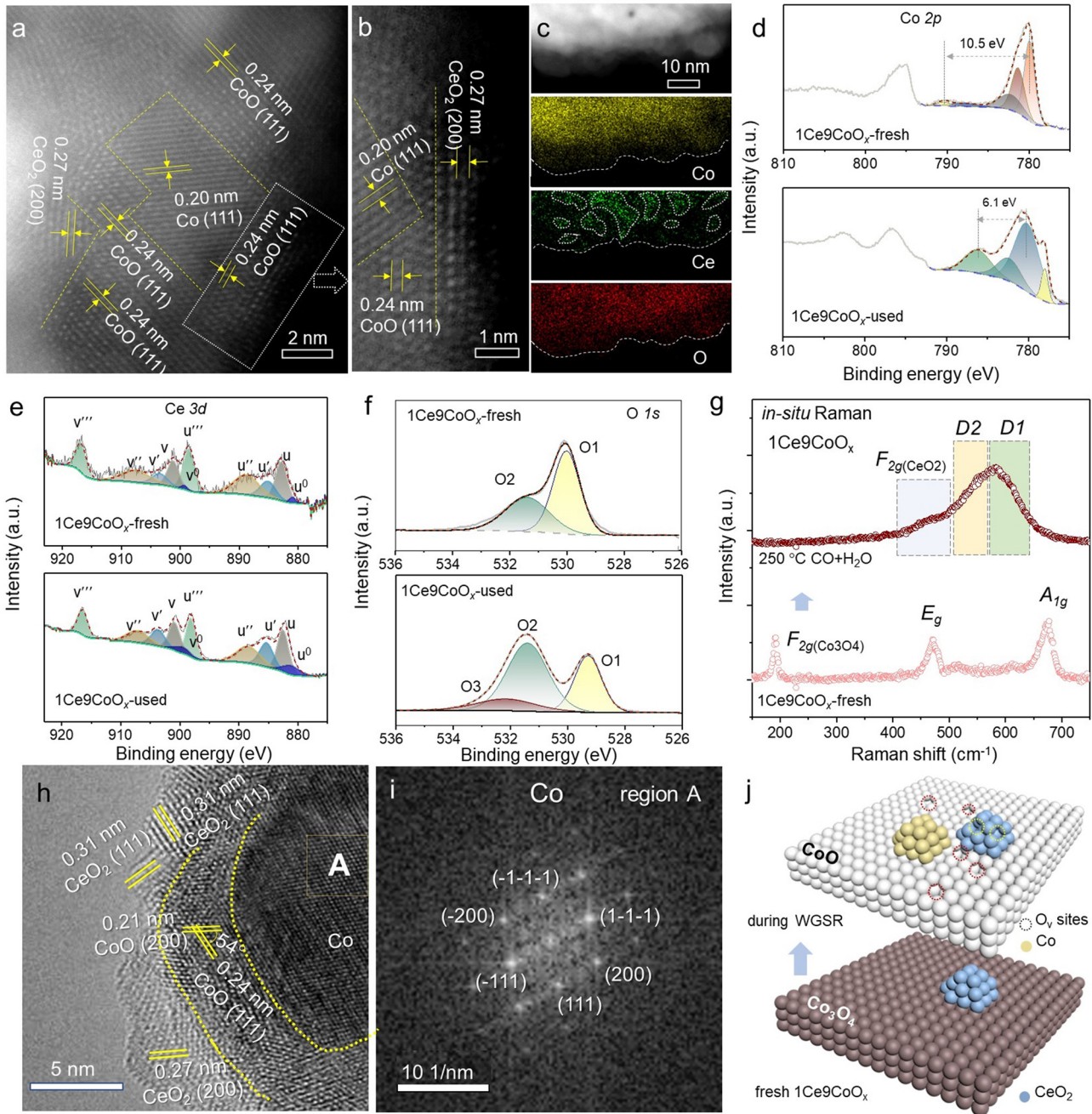

**Fig. 2 | Identification of the reconstructed CeO$_{2-x}$/CoO$_{1-x}$/Co dual-interfaces after WGS reaction. a** HAADF-STEM images; (**b**) magnified STEM image; (**c**) elemental mappings (Co, Ce, and O) for the selected area of 1Ce9CoO$_x$ catalyst after light-off WGS reaction test. **d**–**f** XPS spectra of fresh and used 1Ce9CoO$_x$: (**d**) Co 2*p*, (**e**) Ce 3*d* and (**f**) O 1*s* spectra. The XPS spectra for the spent catalyst were collected after light-off WGS reaction (2%CO/10%H$_2$O/Ar, 180–320 °C). **g** in situ Raman spectra collected under WGS reaction condition at 250 °C. **h** HRTEM image of spent 1Ce9CoO$_x$ catalyst, where the interfacial region is illustrated by dotted line. **i** the corresponding FFT image of selected region A in Fig. 2h. **j** Schematic illustration of the phase transformation within the 1Ce9CoO$_x$ catalyst during the WGS reaction.

microscopy evidence. As shown in Fig. 2h, the crystal lattice spacing ascribed to CeO$_2$(100), CoO(111), and Co(100) could be identified, where the boundary as marked by the dotted line was correspondingly contributed to CoO$_{1-x}$/Co or CeO$_{2-x}$/CoO$_{1-x}$ interfaces. Specifically, the metallic Co species are generally found in the central zone, which is further proven by the corresponding FFT image as shown in Fig. 2i. More importantly, the coexistence of CoO and CeO$_2$ could always be identified in the same regions, such as the regions C as illustrated in Supplementary Figs. 21, 22, where it is adjacent to the metallic Co species in the central area. Similar results could be frequently observed in other HRTEM images, as shown in Supplementary Fig. 23, indicative

of the abundant presence of dual interfaces over the 1Ce9CoO$_x$ catalysts after the WGS reaction. Therefore, on the basis of the above microscopic and spectral evidence, we believed that the initial CeO$_2$-Co$_3$O$_4$ structure re-constructed to the CeO$_{2-x}$/CoO$_{1-x}$/Co dual-interfaces during the WGS reaction (Fig. 2j).

## Functions of CeO$_{2-x}$/CoO$_{1-x}$ and Co/CoO$_{1-x}$ dual-interfaces within the 1Ce9CoO$_x$ catalyst

As an indicator of the heterogenous interaction, the oxygen reducibility of catalysts was determined by the H$_2$ temperature-programmed reduction (H$_2$-TPR) tests[41]. As shown in Fig. 3a, three peaks were

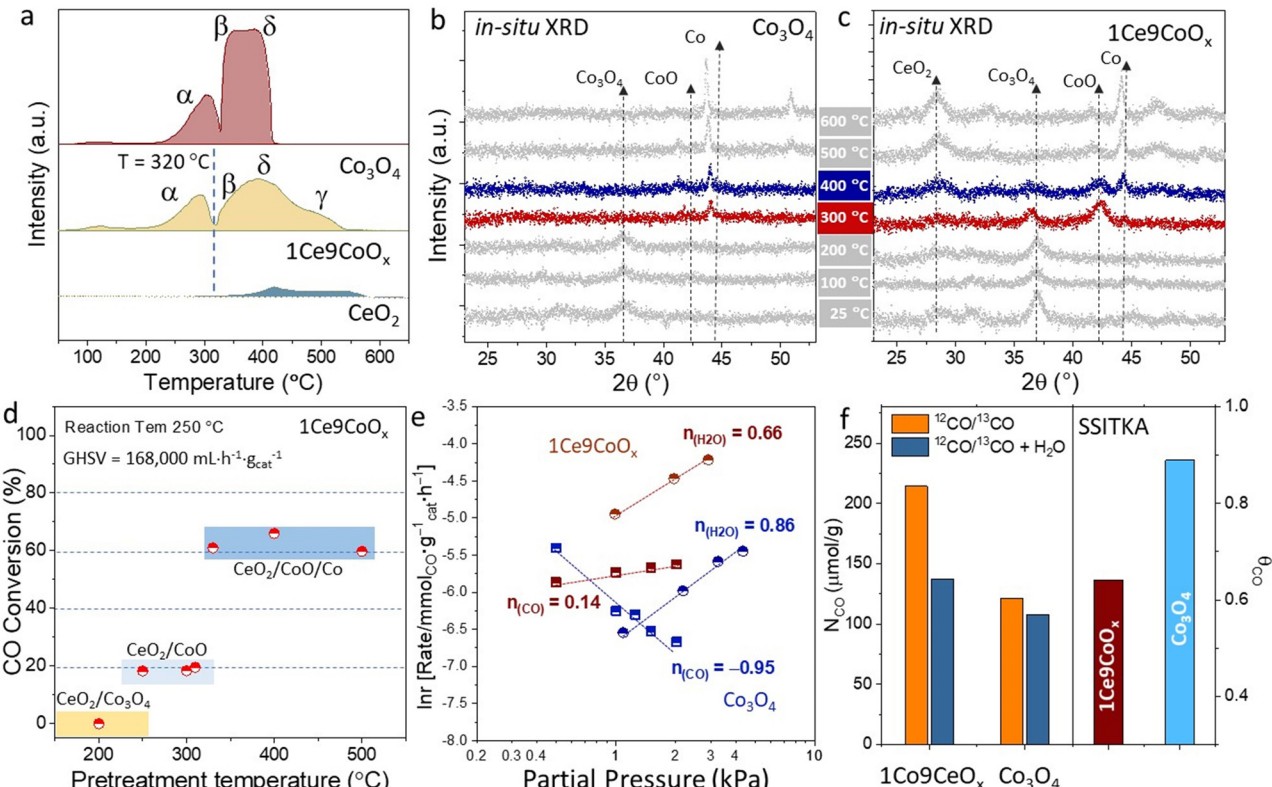

**Fig. 3 | Investigation on the function of the dual interfaces. a** H$_2$-TPR profile of 1Ce9CoO$_x$, Co$_3$O$_4$, and pristine CeO$_2$ samples. **b, c** The in situ XRD patterns were collected under 5%H$_2$/Ar atmosphere with different temperatures for the (**b**) Co$_3$O$_4$ and (**c**) 1Ce9CoO$_x$ catalysts. **d** The CO conversions of WGS reaction at 250 °C for the 1Ce9CoO$_x$ catalysts pretreated by 5%H$_2$/Ar under various temperatures. **e** Apparent reaction order of H$_2$O and CO for the 1Ce9CoO$_x$ and Co$_3$O$_4$ catalysts. **f** The amount of adsorbed CO (N$_{CO}$) and the site coverage of CO ($\theta_{CO}$) determined by the SSITKA results for the 1Ce9CoO$_x$ and Co$_3$O$_4$ catalysts.

respectively identified for pristine Co$_3$O$_4$ at ca. 300, 350, and 410 °C, correspondingly denoted as the α, β and δ peak. In detail, the α reduction peak is indicative of the phase transformation from Co$_3$O$_4$ to CoO; the β and δ peaks imply the reduction process of surface or inner CoO to metallic Co, respectively[15,42]. The ratio of peak area for α/(β+δ) is 1/3.4, which is close to the stoichiometric ratio of 1/3. In comparison with the pristine Co$_3$O$_4$ catalyst, the 1Ce9CoO$_x$ demonstrates a similar α reduction peak at around 290 °C, suggesting the comparable reduction behavior from Co$_3$O$_4$ to CoO for both catalysts. In addition, the relative intensity of β reduction peak was apparently decreased over 1Ce9CoO$_x$; meanwhile, one additional γ peak emerged at an elevated temperature of ca. 500 °C. To note, the α/(β+δ+γ) value of 1/3.2 is close to the stoichiometric ratio of 1/3, implying that the γ peak might be derived from the reduction process of the CoO species strongly interacted with additive CeO$_2$. Videlicet, the surrounded CeO$_{2-x}$ islets can play a vital role in maintaining the O-containing structure of Co species even at relatively high temperatures (>400 °C) under a reductive atmosphere. The ability of small CeO$_2$ nanoparticles to provide O atoms for oxidizing adjacent Co atoms was also validated by the auto-regenerated α peak as shown in the second-round TPR profile (Supplementary Fig. 24). These results demonstrated the pivotal role of CeO$_{2-x}$ islets in fabricating the CeO$_{2-x}$/CoO$_{1-x}$/Co dual interface.

The in situ XRD patterns collected under the H$_2$ atmosphere were further used to substantiate the phase evolution over the 1Ce9CoO$_x$ and Co$_3$O$_4$ catalysts. We observed a distinct phase transformation from Co$_3$O$_4$ to metallic Co over the pristine Co$_3$O$_4$ catalysts at 300 °C (Fig. 3b), coupling with negligible signal centered at 42.3° for the CoO phase. In sharp contrast, a remarkable diffracted peak at 42.3° for CoO emerged at 300 °C over the 1Ce9CoO$_x$ sample, while the signal for

metallic Co is relatively weak at around 44.4°. This evidence is well consistent with the H$_2$-TPR results that the small CeO$_{2-x}$ nanoparticles can maintain the oxidized state of Co species with a broader temperature range under reductive conditions, which is in line with its stabilization effect found in the previous report[43].

To validate the superiority of CeO$_{2-x}$/CoO$_{1-x}$/Co dual-interfaces, pretreatments with 5%/H$_2$/Ar under various temperatures were conducted on the 1Ce9CoO$_x$ to differentiate the initial phase state of catalysts (Fig. 3c and Supplementary Table 2), where the samples labeled by 1Ce9CoO$_x$-x respectively (x is pretreated temperatures). After pre-reducing at 200 °C (1Ce9CoO$_x$-200), the Co$_3$O$_4$ phase was predominant according to the in situ XRD results, and the 1Ce9CoO$_x$-200 was almost inactive for catalyzing the WGS reaction. As the temperature elevated to 250 °C, the initial CO conversion was boosted to ~20%, indicating that the CoO site is superior to Co$_3$O$_4$. Interestingly, compared with the 1Ce9CoO$_x$-350 sample, the slightly lower reduction temperature of 320 °C led to a significant decrease in CO conversion from ~60% to ~20% for the 1Ce9CoO$_x$-320. This inflected temperature of 320 °C coincides nicely with the starting temperature of the β reduction process (Co$^{2+}$ → Co$^0$) as depicted in the H$_2$-TPR profile (Fig. 3a), indicating that metallic Co$^0$ species is an essential component for the efficient catalysis process. Remarkably, physically mixed metallic Co catalysts and CeO$_2$ nanoparticles displayed an inferior CO conversion of 5.2% at 250 °C (Supplementary Table 1), proving that CoO sites induced by coordinated CeO$_{2-x}$/CoO$_{1-x}$ interaction were also indispensable for highly active performance. Furthermore, the copresence of CoO and metallic Co coupling with best catalytic performance (Fig. 3d) was observed after in situ reduction under 5%H$_2$/Ar at 400 °C for the 1Ce9CoO$_x$, reconfirming the co-existing of dual active Co sites (Co$^0$ and Co$^{2+}$) was crucial for efficiently catalyzing WGS reaction.

The apparent reaction orders of $H_2O$ and CO were determined at ca. 230 °C over the 1Ce9CoO$_x$ and Co$_3$O$_4$ catalysts. The $H_2O$ order on 1Ce9CoO$_x$ is relatively lower than the Co$_3$O$_4$ catalyst (0.66 vs. 0.84, Fig. 3e), implying that the CeO$_{2-x}$/CoO$_{1-x}$/Co dual-interfaces have more substantial adsorption and coverage of O/OH/$H_2O$ species than bare metallic Co surface. The CO order is 0.14 on the 1Ce9CoO$_x$ catalyst, whereas the CO order for the Co$_3$O$_4$ catalyst is −0.95. The inverse-first order of CO upon bare Co$^0$ sites within Co$_3$O$_4$ proved the excessively strong CO adsorption at relatively low temperatures[17,23], which might correspondingly block the active sites for the $H_2O$ activation. The moderate CO adsorption ability for the 1Ce9CoO$_x$ suggested that the CoO$_{1-x}$/Co interface behaved differently from its metallic counterpart[23]. We further substantiate this speculation by conducting steady-state isotopic transient kinetic analysis (SSITKA) experiments over the 1Ce9CoO$_x$ and Co$_3$O$_4$ catalysts. By in situ switching $^{12}$CO/He to $^{13}$CO/Ar with or without feeding 3%$H_2O$ inside, the total number of active sites ($N_{tot}$) and adsorbed amount of reversibly adsorbed CO ($N_{CO}$) molecules under steady state were determined as shown in Supplementary Fig. 25[44]. Subsequently, dividing the $N_{CO}$ with $N_{tot}$ gave the surface coverage of CO ($\theta_{CO}$) with the presence of $H_2O$ molecules. As shown in Fig. 3f, the metallic Co$^0$ sites within the Co$_3$O$_4$ catalyst demonstrated a higher $\theta_{CO}$ of 0.89 in comparison with 0.64 for the

CoO$_{1-x}$/Co interface within 1Ce9CoO$_x$ catalyst at 250 °C, further proving that the excessively stable CO molecules exactly impaired $H_2O$ activation upon the pristine surface of metallic Co$^0$. Therefore, the reconstructed CoO$_{1-x}$/Co interface effectively weakened the CO poisoning effect at the relatively low reaction temperature, which provides more active sites for the following surface reaction.

## Synergistic reaction pathway proceeded upon CeO$_{2-x}$/CoO$_{1-x}$ and CoO$_{1-x}$/Co dual-interfaces

In situ Raman spectra were sequentially collected under different modes over the 1Ce9CoO$_x$ catalyst to offer a deeper insight into the detailed reaction pathway. Figure 4a depicted that numerous V$_{(CoO)}$ and V$_{(CeO2)}$ with defect modes at ca. 540 and 580 cm$^{-1}$ were generated in the interface region over 1Ce9CoO$_x$ catalyst when pretreating the 1Ce9CoO$_x$ catalyst with 5%$H_2$/Ar at 400 °C. Based on previous reports, the value of the $D/F_{2g}$ ratio can qualitatively represent the concentration of O$_{v(CeO2)}$[37]. Notably, the $D_{(CeO2)}/F_{2g}$ ratio of ~2.3 over 1Ce9CoO$_x$ catalyst is much higher than that of the pristine CeO$_2$ nanoparticles with an averaged diameter of ca. 5−8 nm ($D/F_{2g}$ = ~0.1, Supplementary Fig. 26), indicative of the vastly boosted concentration of oxygen vacancy by forming CeO$_{2-x}$/CoO$_{1-x}$ interface. When co-feeding CO and $H_2O$ inside, the relative intensity of $D$ mode signal at 540 cm$^{-1}$

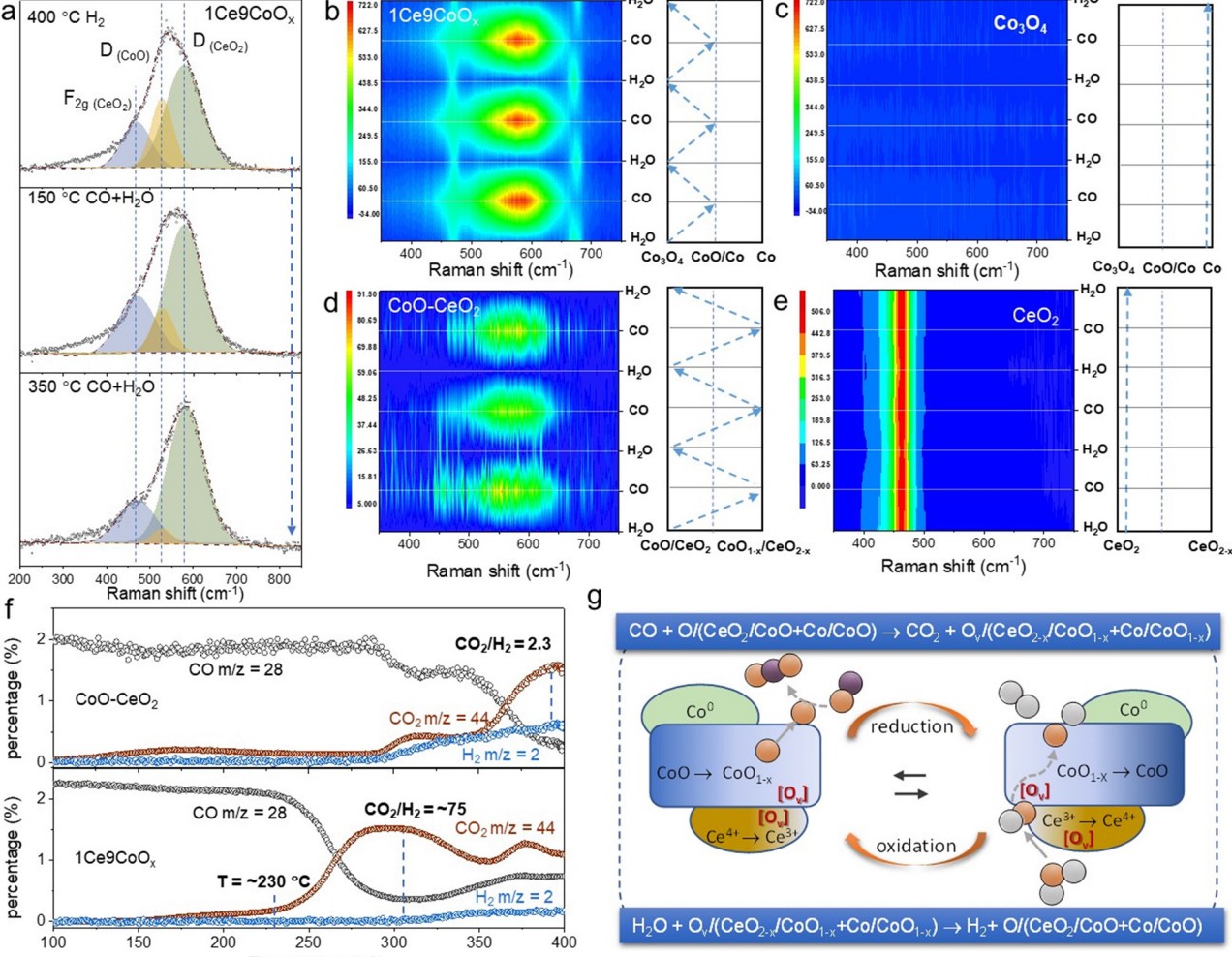

**Fig. 4 | Study of the synergistic mechanism on the dual-interfaces. a** In situ Raman spectra collected under 5%$H_2$/Ar or 2%CO/3%$H_2O$/Ar atmospheres at 400 °C. **b–e** In situ Raman spectra continuously collected as the feeding gas switched between 3%$H_2O$/Ar and 5%CO/Ar at 250 °C for (**b**) 1Ce9CoO$_x$, (**c**) Co$_3$O$_4$, (**d**) CoO-CeO$_2$ and (**e**) CeO$_2$ catalysts. The right part demonstrates the transformations of Co and CeO$_2$ species during experiments. **f** CO-TPSR profile for the CoO-CeO$_2$ catalyst and 1Ce9CoO$_x$ catalyst. The catalysts were pre-hydroxylated (3%$H_2O$/Ar, 250 °C) before experiments. **g** Schematic illustration of plausible WGS reaction pathway upon the CeO$_{2-x}$/CoO$_{1-x}$/Co dual interface.

apparently decreased at 150 °C and further weakened at 350 °C (Fig. 4a), suggesting that these vacancy structures participated in dissociating $H_2O$ molecules and can be refilled by the dissociated O(H) species at elevated temperatures.

Next, the CO-$H_2O$ switch test was used to visualize the stepwise process between the $CeO_{2-x}$/$CoO_{1-x}$ and $CoO_{1-x}$/Co interfaces for the $1Ce9CoO_x$ catalyst (Fig. 4b). After feeding 3%$H_2O$/Ar onto the surface of $1Ce9CoO_x$ catalyst at 250 °C, typical $E_g$ mode at 480 cm$^{-1}$ and $A_{1g}$ mode at 680 cm$^{-1}$ for crystallized $Co_3O_4$ were observed, while the Raman signals at 540 and 580 cm$^{-1}$ for $V_{(CoO)}$ and $V_{(CeO2)}$ vanished. This experimental fact evidenced that the dissociated O(H) species from $H_2O$ molecules can refill the vacant structures and oxidize the $CeO_{2-x}$/$CoO_{1-x}$ interface at a relatively low temperature. Subsequently, the $D$ mode signals at 540 and 580 cm$^{-1}$ for the $V_{(CoO)}$ and $V_{(CeO2)}$ recurred when feeding 5%CO/Ar inside, implying the reversible regeneration of vacancies and consumption of active O(H) species at the $CeO_{2-x}$/$CoO_{1-x}$ interface. In sharp contrast, no detectable structure transformations were observed upon the pristine surface of metallic Co species during the continuous CO-$H_2O$ switching tests for the $Co_3O_4$ catalyst (Fig. 4c), implying that the presence of $CeO_2$ is essential for boosting the $H_2O$ activation ability.

The CoO-$CeO_2$ catalyst and bare $CeO_2$ nanoparticles were prepared (Supplementary Figs. 27–29) and investigated further to dig out the role of the $CoO/CeO_2$ interface. Typical Raman active signals for $CeO_2$ centered at 463 cm$^{-1}$ ($F_{2g}$ mode) and ca. 580 cm$^{-1}$ ($D$ mode) could be identified during the CO-$H_2O$ sequential switching tests (Fig. 4d). In the presence of CO molecules at 250 °C, the signal of $D$ mode could be enhanced over the CoO-$CeO_2$ catalyst, implying that the O species occupied vacancies could be easily removed in this step. More importantly, this consumption of O species upon the vacancies could be regenerated as $H_2O$ molecules purge inside. However, the bare $CeO_2$ demonstrated negligible variation for the $D$ mode at 580 cm$^{-1}$ as CO or $H_2O$ feeding inside (Fig. 4e), proving the low O species activity in the absence of the $CoO/CeO_2$ interface. This recyclability of O vacancies for the CoO-$CeO_2$ catalyst induced by gas switching between reactant molecules was indicative that the structural changes of $CoO/CeO_2$-$CoO_{1-x}$/$CeO_{2-x}$ might participate in the CO oxidation step and $H_2O$ dissociation step.

Additionally, it should be noted that the reversible structural changes over the CoO-$CeO_2$ catalyst were different from those of the $1Ce9CoO_x$ catalyst. As shown in Fig. 4b, typical phonon vibration signals at 480 cm$^{-1}$ ($E_g$ mode) and 680 cm$^{-1}$ ($A_{1g}$ mode) for $Co_3O_4$ were observed as the $H_2O$ molecules fed inside, while it was absent for the CoO-$CeO_2$ catalyst (Fig. 4c). This implied that the oxidation behavior of the Co species should be strongly correlated with the presence of the $Co/CoO$ interface over the $1Ce9CoO_x$ catalyst. Videlicet, the $H_2O$ molecules could dissociate on the $CoO/CeO_2$ interface, while the dissociated O-related species were unable further to oxidize cobalt species without the $Co/CoO$ interface. In addition, we collected the $H_2$-TPR profile for the $1Ce9CoO_x$ catalysts after the in situ WGS reaction (250C, 2%CO/3%$H_2O$/Ar). Interestingly, as shown in supplementary Fig. 30, another sharp peak centered at 172 °C was observed when the $1Ce9CoO_x$ was exposed to the feed gas, which might belong to the reduction of active O species generated by redox reaction cycles on the surface. As expected, this deduction was also supported by the CO-TPSR results. In the typical carboxyl pathway, the adsorbed CO molecules would react with the dissociated OH* group by forming the HOCO intermediate upon the surface of catalysts[31,45]. Based on the prerequisite that adsorbed molecules reacted on the surface of the catalyst as $CO + OH \rightarrow CO_2 + 1/2H_2$[26], the CO temperature-programmed surface reaction (CO-TPSR) test normally can validate such an association process from the stoichiometric ratio (1/2) for produced $H_2$ and $CO_2$. As shown in Fig. 4f, the generated products for the CoO-$CeO_2$ catalyst follow a $CO_2$/$H_2$ ratio of ~2/1, implying that the O-related species involved in the surface reaction should be hydroxy groups

rather than active O atoms. However, when the Co/CoO interface was generated by pretreatment at 400 °C (Fig. 3c), the $1Ce9CoO_x$ catalyst with dual-interface structure demonstrated a distinctly different $CO_2$/$H_2$ evolution during the CO-TPSR test. Negligible $H_2$ ($CO_2$/$H_2$ » 2) was identified in the CO-TPSR profile over the hydroxylated $1Ce9CoO_x$ catalyst, indicating that the CO molecules on $CoO_{1-x}$/Co interface might be oxidized by activated O atoms rather than OH groups upon the dual-interfaces. Moreover, the evolution slope of produced $CO_2$ in CO-TPSR (m/z = 44, Fig. 4f) was well in step with the evolution of CO conversions as shown in Fig. 1b, in which the reaction activity increased sharply at about 230 °C. Such a correlation rationalized that the surface reaction between activated O atoms and CO molecules contributes primarily to the WGS reaction activity over the $1Ce9CoO_x$ catalyst. Based on these results, it is reasonable to conclude that the Co/CoO interface within the $1Ce9CoO_x$ catalyst was indispensable for dissociating $H_2O$ molecules to active O atoms.

The H/D kinetic isotopic effect (KIE) has been widely used to explore the WGS reaction mechanism[46], owing to differentiated zero-point energies of the D- and H- labeled molecular species induced by the significant mass distinction between the D and H atoms. The relatively larger H/D KIE value of 5.6 indicated the O-D bond cleavage involved in the primary kinetic mode on the bare $Co^0$ surface (Supplementary Fig. 31), implying that the breaking O-H in $H_2O$ molecules should be the rate-determining step (RDS) of WGS reaction process over the surface of the pristine Co. In contrast, the $1Ce9CoO_x$ catalyst demonstrated a lower KIE value of 1.9, implying a normal isotopic effect on the $CeO_{2-x}$/$CoO_{1-x}$ dual interface structure. Considering its $H_2$ reaction order of −0.9 (Supplementary Fig. 32), we speculated that a slow H transfer/spillover step might be involved in the reaction pathway $1Ce9CoO_x$ catalyst according to previous research findings[46].

Based on these observations mentioned above, as schematically illustrated in Fig. 4g, we propose that two sequentially redox steps were synergistically involved over the $CeO_{2-x}$/$CoO_{1-x}$/Co dual-interfaces while catalyzing the WGS reaction:

$$CO + O/(CeO_2/CoO + Co/CoO) \rightarrow CO_2 + O_v/(CeO_{2-x}/CoO_{1-x} + Co/CoO_{1-x})$$
(1)

$$H_2O + O_v/(CeO_{2-x}/CoO_{1-x} + Co/CoO_{1-x}) \rightarrow H_2 + O/(CeO_2/CoO + Co/CoO)$$
(2)

To verify the rationality of the proposed mechanism, we conducted DFT calculations to investigate the reaction path on the dual-interface structures. Since the above experimental data provided a strong indication that CoO sites modified the CO adsorption and $CeO_{2-x}$ islets contribute to efficient $H_2O$ activation, the reaction energy of elementary steps over $Co^0$ sites or $CeO_{2-x}$/CoO interfacial sites was investigated. On the basis of the experimental characterization findings, we conducted three catalyst models as illustrated in Supplementary Fig. 33 (Co(0001), $CeO_{2-x}$/CoO(100) and $Co_{10}$/CoO(100)). The CO adsorption energy was determined to be −1.69 eV on the pristine Co(0001), which would thermodynamically compete for the $H_2O$ dissociation process on the same $Co^0$ sites (Supplementary Fig. 34). The intense CO adsorption on Co(0001) surface led to the poisoning of active sites and induced sequential high activation energy of 1.15 eV and 1.08 eV for forming *COOH intermediate and subsequently generating *$CO_2$ and *$H_2$ on Co(0001) surface, implying the detrimental effect of strong CO adsorption on independent $Co^0$ sites. In contrast, the $CeO_{2-x}$/CoO interface showed moderate energy of −0.28 eV via the CO adsorption process upon $Co^{2+}$ sites, and coupling with a reasonably low reaction barrier of 0.89 eV (TS1) to subsequently react with lattice O atom of $CeO_{2-x}$ (Fig. 5a, b), resulting in the generation of an $O_{v(CeO2-x)}$

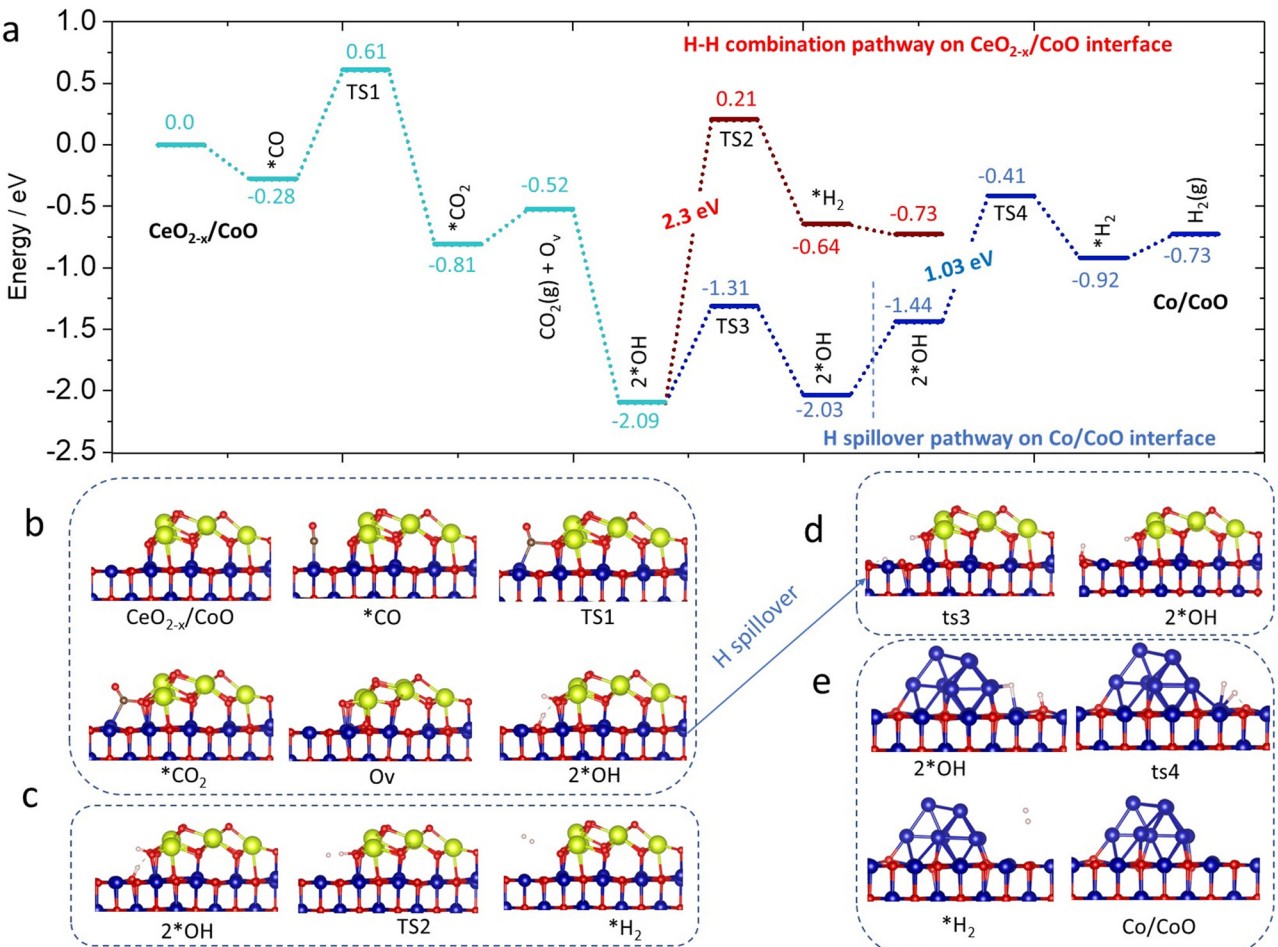

**Fig. 5 | Reaction pathway and corresponding optimized structures for the water gas shift reaction over dual interfaces. a** Energy profile for the CO oxidation with lattice O, $H_2O$ dissociation and $H_2$ formation steps on the $CeO_{2-x}$/CoO(100) and $Co_{10}$/CoO(100) interfaces, respectively. The $x$-axis shows the reaction intermediates and transition states (TS); the $y$-axis demonstrates the relative energy. **b** Structures of the CO molecules react with interfacial O from $CeO_{2-x}$. **c** Structures of the $^*H_2$ derived through an H-H direct combination way on the $CeO_{2-x}$/CoO interface. **d** Illustrations of H diffusion on CoO(100) surface. **e** Illustrations of $H_2$ formation at the $Co_{10}$/CoO(100) interface. Co, Ce, O, and H atoms are shown in purple, yellow, red and white, respectively.

vacancy. As-formed $O_{v(CeO2-x)}$ within the $CeO_{2-x}$/CoO interface enables $H_2O$ molecules to dissociate into two $^*OH$ with a favorable exothermic step of 1.28 eV, which is consistent with the superior activation ability of $1Ce9CoO_x$ catalyst as experimentally evidenced above.

To note, the metallic $Co^0$ atoms are also prerequisite sites for high WGS activity according to experimental results shown in Fig. 3, whereas its exact function is still ambiguous in previous reports[15,18,47]. In addition, our DFT calculation results demonstrated that the downstream step of $H_2$ direct formation from two independent $^*OH$ groups at $CeO_{2-x}$/CoO interface underwent the transition state of TS2 (Fig. 5c), which had an extremely high barrier of 2.3 eV (red line). A similar process was also found in calculation results for the $Ir_1O_x$/$FeO_x$ catalyst that direct formation of $^*H_2$ molecules from two independent H atoms revealed a disadvantageous energy barrier of 3.25 eV[5]. The enormous energy barrier implied that the single $CeO_{2-x}$/CoO interface is insufficient for a thermodynamically favored WGS reaction cycle. Comprehensively, in consideration of the indispensable role of metallic $Co^0$ sites for an efficient catalysis process, we calculated the $H_2$ formation step at the $Co_{10}$/CoO(100) interface. As expected, the H atom diffuses on CoO(100) with a mild migration barrier of 0.78 eV (Fig. 5d). In addition, it is interesting that breaking the H atom from OH onto CoO(100) with the aid of the metal $Co^0$ site is relatively facile with a rational energy barrier of 1.03 eV (Fig. 5e). The transition structure

(TS4) comprises $H^{δ+}$ and $H^{δ-}$ atoms that are coordinated with the O atom and $Co^0$ atom, respectively (Supplementary Fig. 35). After two H atoms approach each other, the $H_2$ molecule is formed and only has physical adsorption at the $Co^0$/CoO(100) interface, which could easily be removed. We underlined the synergistic participation of dual interfaces, $CeO_{2-x}$/CoO and $Co^0$/CoO, in the proposed redox mechanism, which well rationalized our experimental finding as discussed above.

In summary, a highly active dual-interfacial catalyst was fabricated through a spray pyrolysis approach, demonstrating boosted WGS reaction activity than other Co-based catalysts and even comparable with some noble catalysts. We have conducted a combined experimental and theoretical study to explore the anthemically active structure and reaction pathway over the dual-interfacial structure. The $CeO_{2-x}$/$CoO_{1-x}$/Co dual-interfaces are structurally active for efficiently catalyzing WGS reaction, where the surrounded presence of small $CeO_{2-x}$ islets is the induction for dominantly existing $CoO_{1-x}$ in the intermediate region. The as-formed $CoO_{1-x}$ can optimize the CO activation step by avoiding CO poisoning due to the excessively strong CO-$Co^0$ binding. A redox mechanism comprising dual interfaces ($CeO_{2-x}$/$CoO_{1-x}$ and Co/$CoO_{1-x}$) synergistically participating in a reaction cycle has been identified. The $CeO_{2-x}$/$CoO_{1-x}$ interface offers an activated O atom to react with adsorbed CO molecule and also takes part in initially dissociating $H_2O$ into two adsorbed OH groups. Directly breaking two

dissociated OH groups to form an $H_2$ molecule is energetically unfavorable at the $CeO_{2-x}/CoO_{1-x}$ interface, while the H atoms are much easier to migrate and desorb at the $Co/CoO_{1-x}$ interface.

## Methods

### Catalyst preparations

The $CeCoO_x$ mesoporous hollow spheres were synthesized by a spray pyrolysis method[48]. 8 mmol of mixed cobalt nitrate and cerium nitrate (molar ratio of Ce/Co is 1/9, 9/1, and 0/10) were added to the 60 mL ethanol. After 10 min stirring, the stock solution was transferred to the household ultrasonic humidifier to produce microspheres via the aerosol-spraying process. The atomized spray was sequentially carried by pure $N_2$ flow through a 90 cm -length glass tube placed in a tube furnace at 400 °C. Then, the hybrid powders were collected and further dried in an oven at 70 °C overnight. The reference catalyst of $1Al9CoO_x$ was prepared with the same procedures described above, in which the molar ratio of Al/Co is 1/9.

### Hydrogen temperature programmed reduction ($H_2$-TPR)

The $H_2$-TPR tests were conducted in a Builder PCSA-1000 instrument (Beijing, China) equipped with a thermal conductivity detector (TCD) to detect $H_2$ consumption. The fresh catalysts (ca. 30 mg, 20–40 mesh) were pretreated at 300 °C in the air (0.5 h). Following cooling to room temperature, the sample was flushed using pure $N_2$ at room temperature for about 30 min, then switched to 10% $H_2$/Ar and heated from room temperature to 900 °C with a ramping rate of 5 °C·$min^{-1}$. The second-round $H_2$-TPR profile for the catalyst was continuously collected after conducting the first-round test without exposure to air.

### X-ray diffraction (XRD)

The ex-situ XRD patterns were obtained by a PANalytical X'pert3 powder diffractometer (40 kV, 40 mA, $\lambda_{Cu-K\alpha}$ = 0.15418 nm) with an acquisition time of 8.5 min in the range of 10–90°. The in situ XRD patterns were obtained from the same machine with an Anton Paar XRK-900 reaction chamber. Samples were loaded in a ceramic sample holder (diameter of 10 mm; depth of 1 mm) and then treated with various conditions. The in situ reaction camber was heated from room temperature to 600 °C (interval: 100 °C) with a ramping rate of 30 °C/min under 5%$H_2$/Ar (30 mL/min). Two rounds of measurements lasting for 20 min were carried out for each selected temperature. The second measurement round was collected and used to determine the structure of the catalysts.

### Raman test

The ex-situ and in situ Raman spectra were recorded on a LabRAM HR800 spectrometer (HORLBA Jobin Yvon) using 473 nm laser excitation. The spectral resolution was 2 $cm^{-1}$ with a scanning range of 200 to 800 $cm^{-1}$. The micro-Raman in situ reactor (Xiamen TOPS) with a thermo-module was used for in situ experiments. Before each measurement, ca. 25 mg of the catalysts were pretreated in 5%$H_2$/Ar (99.997%, 30 mL/min) at 400 °C for 60 min. Then, the in situ Raman spectra were collected in the following two condition modes: (1) ~3% $H_2O$/Ar or 5%CO/Ar with a flow rate of 30 mL/min alternatively flowed inside the reaction chamber at 250 °C; (2) the WGS reaction gas (5%CO/~3%$H_2O$/Ar, 30 mL/min) was continuously injected to the reaction cell at various temperatures. The six-way valve was used to avoid the plausible oxidation of the catalysts.

### X-ray photoelectron spectroscopy (XPS)

X-ray photoelectron spectroscopy (XPS) analysis was carried out on an Axis Ultra X-ray photoelectron spectrometer using Al Kαradiations with the C $1s$ line at 284.8 eV calibrating the binding energies. Quasi in situ XPS experiments were conducted on a Thermo Scientifi-cESCALAB Xi+ XPS instrument.

### Catalytic tests

A fixed-bed flow reactor was used for the WGS reaction tests on the catalysts. The reaction gas was 2%CO/10%$H_2O$/$N_2$ (99.997% purity), and the total GHSV was 42,000 or 168,000 mL $g_{cat}^{-1}$ $h^{-1}$. The catalyst was pretreated in synthetic air at 300 °C for 30 min before the measurement. The reactivity results of catalysts were measured from 150 to 300 °C per step under reaction gas. A non-dispersive IR spectroscopy was used to quantify the outlet gases online. The residual water was gathered in an ice trap before the IR gas analyzer.

## Data availability

The main data supporting the findings of this study are available within the paper and its Supplementary Information. Additional data are available from the corresponding authors upon reasonable request. Source data are provided with this paper.

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

## Acknowledgements

This work is funded by the National Key Research and Development Program of China (2021YFA1501103), the National Science Fund for Distinguished Young Scholars of China (22225110), the National Science Foundation of China (22075166, 22271177), the Young Scholars Program of Shandong University. We thank the Center of Structural Character-izations and Property Measurements at Shandong University for the help on sample characterizations.

## Author contributions

C.-J.J. supervised the work; X.-P.F., C.-P.W. and C.-J.J. had the idea and designed the experiments. X.-P.F. and C.-J.J. analyzed the results and wrote the manuscript; X.-P.F. and C.-P.W. performed the catalytic tests, H2-TPR and TPSR tests. X.-P.F. and W.-W.W. performed the in situ in situ XRD, in situ Raman and quasi in situ XPS; X.-P.F., Z.J., and C.-P.W. designed and prepared the catalysts; J.-C.L. performed the DFT calcu-lation; C.M. performed the aberration-corrected HAADF-STEM mea-surements and analyzed the results. All authors have given approval to the final version of the manuscript.

## Competing interests

The authors declare no competing interest.
