## [Peer Review File · Nature Communications]

REVIEWER COMMENTS

Reviewer #1 (Remarks to the Author):

The authors have tried to find out the mechanism over $\text{CeO}_{2-x}/\text{CoO}_{1-x}/\text{Co}$ catalyst with dual-interface structure for WGS reaction. This paper can be published after revisions.

1. In Fig. 1d, the feed gas used in the stability test contains CO_2 and H_2 , which is different from the feed gas used in the catalytic test (Fig. 1b). Thus, the authors need to explain the reasons.

2. The authors claimed that highly active dual-interfaces of $\text{CeO}_{2-x}/\text{CoO}_{1-x}/\text{Co}$ are constructed during the WGS reaction. The co-existence of metallic Co and CoO was fully demonstrated by the HAADF-STEM, XRD, and XPS results. However, it is hard to confirm the structure of dual interfaces. Dual active Co sites (Co^0 and Co^{2+}) can be dispersed on CeCoO_x mesoporous hollow spheres. The authors should provide detailed explanation.

3. In this study, each in-situ characterization of prepared catalysts was conducted at different WGS reaction conditions (temperature and feed gas composition). The authors need to conduct the characterizations with the same condition of in-situ WGS reaction.

4. The relevant paper should be cited in the reference.

Target-oriented water–gas shift reactions with customized reaction conditions and catalysts

Chemical Engineering Journal, 2023, 458, 141422

Reviewer #2 (Remarks to the Author):

Fu et al. studied a complex Co-ceria system for the water gas shift reaction. The developed $1\text{Ce}_9\text{CoO}_x$ catalyst shows high activity and exhibits different structure changes compared with

reference catalysts. These results suggest that 1Ce9CoO_x possesses certain structural features that promotes the reaction. To characterize the structure of catalysts, multiple techniques including XRD, TEM, XPS and Raman were used. However, I think this part needs to be improved. Currently, the set of tools reveals defects and possible Co species existing in the systems but there is no strong evidence connecting defects and different interfaces and how they participate into the reaction. Even for the interfaces, only based on TEM, it is hard to tell what interfaces exist in the system. Therefore, it is hard to claim that the improved activity is due to the synergistic effects of proposed dual interfaces. Other than these, here are additional questions:

1. XRD has limitations in detecting species with small sizes or with disordered structures. Need to provide additional evidence to show that in the fresh 1Ce9CoO_x, Co is in only in Co₃O₄ phase.
2. It is hard to tell the size of CoO_x and CeO_x species by using STEM. In addition, STEM is a local technique. Suggest to performing XAS measurements to provide complementary results.
3. Could you please compare the Ce and Co XPS spectra collected before and after the reaction? Now Fig. 2d,e only show the spectra collected after the reaction. Without comparison, one cannot tell how Co and Ce species change due to the reaction.
4. From Fig.2d, it is hard to tell the existence of Co metallic. Could you please do the fitting to confirm its existence?
5. Fig. S12b also plotted the Co 2p XPS of used 1Ce9CoO_x but the spectral features are different from those in Fig. 2d. Please examine.
6. In XPS, the observed Co²⁺ is assigned to Co²⁺ in CoO or CoO_{1-x}. Is it possible that Co²⁺ is from the interface of Co/CeO₂?
7. I don't think one can use the atomic ratio of O/Co to claim that Co is in CoO_{1-x}. XPS detected not only Co²⁺ but also Co⁰. The ratio of O/Co must be smaller than 1. Besides, if Co is CoO_{1-x}, the oxidation state of Co would be different from Co²⁺, will you see peak shift in XPS?
8. Page 5, "The amount of the CeO₂-Co interfacial sites within the 1Ce9CoO_x catalyst was defined by the perimeter outline of small CeO_{2-x} particles with favored thermostability, which consequently resist the loss of active interfacial Co-CeO₂ sites under reaction condition." Page 7, "The differentiated O/Co ratios in the surface region of catalysts acquired over quasi in-situ experiments clearly showed that the active phase is nonstoichiometric cobalt monoxide (CoO_{1-x}) rather than CoO, ..." How do you know these species or sites are active? Because they exist in the catalyst?
9. What is the evidence for the formation of Co/CoO_{1-x} interface?

Responses to the reviewers

We would like to thank the editors and the reviewers for their valuable comments. We have addressed all comments, as detailed below. All the changes performed in the text of the manuscript or the SI are highlighted in yellow.

To Reviewer #1:

Comment 1. In Fig. 1d, the feed gas used in the stability test contains CO₂ and H₂, which is different from the feed gas used in the catalytic test (Fig. 1b). Thus, the authors need to explain the reasons.

Response: Thanks for the reviewer's valuable suggestion. For better illustration, a harsh reaction condition was selected to differentiate the catalytic performance of various catalysts in the manuscript (**Fig. 1d**, 5%CO/15%H₂O/5%H₂/5%CO₂/N₂, GHSV = 400,000 mL·g_{cat}⁻¹·h⁻¹), which is closer to the realistic atmosphere environment. In addition, we also supplementarily conducted the stability tests for the catalysts with the same atmosphere used in **Fig. 1b** (2%CO/10%H₂O/N₂), as shown in **Figure R1-1**. Similarly, the 1Ce9CoO_x catalyst performed much better than the bare Co₃O₄ catalyst with a time on stream of 1800 min. Thanks for the reviewer's valuable advice again. **We added the corresponding description and figure in the revised manuscript (Supplementary Fig. 7, page 4, lines 7-9; page 5, lines 1-4).**

Figure R1-1. Long-term WGS reaction test over the 1Ce9CoO_x and Co₃O₄ catalysts at 250 °C. The other Reaction condition: 2%CO/10%H₂O/N₂, GHSV = 168,000 mL/g_{cat}/h.

Comment 2. The authors claimed that highly active dual-interfaces of $\text{CeO}_{2-x}/\text{CoO}_{1-x}/\text{Co}$ are constructed during the WGS reaction. The co-existence of metallic Co and CoO was fully demonstrated by the HAADF-STEM, XRD, and XPS results. However, it is hard to confirm the structure of dual interfaces. Dual active Co sites (Co^0 and Co^{2+}) can be dispersed on CeCoO_x mesoporous hollow spheres. The authors should provide detailed explanation.

Response: Thanks for the valuable suggestion. To validate the structure of the dual interface within the $\text{CeO}_{2-x}/\text{CoO}_{1-x}/\text{Co}$ catalyst as proposed in the manuscript, we conducted experiments to clarify it from two aspects: confirming the presence of $\text{CeO}_{2-x}/\text{CoO}_{1-x}$ and $\text{CoO}_{1-x}/\text{Co}$ interface through the characterization results of spent catalysts; illustrating the functional cooperativity of each interface through comprehensive comparison over various reference catalysts.

Figure R1-2. (a-d) Atomic-resolution HAADF-STEM images of the selected area of $1\text{Ce}_9\text{CoO}_x$ after transient WGS reaction to determine the microstructure of spent catalysts. The CoO nanoparticles are surrounded by a few small CeO_2 nanoparticles with a diameter of about 2–5 nm.

We approve the reviewer's point and further supplement microscopy evidence for the presence of $\text{CeO}_{2-x}/\text{CoO}_{1-x}$ and $\text{CoO}_{1-x}/\text{Co}$ interface within the spent catalysts. As shown in **Figure R1-2**, the crystal lattice ascribed to $\text{CeO}_2(100)$, $\text{CoO}(111)$ and $\text{Co}(100)$ could be identified, where the edged regions, as marked by the dotted line, were

correspondingly contributed to $\text{CoO}_{1-x}/\text{Co}$ or $\text{CeO}_{2-x}/\text{CoO}_{1-x}$ interfaces. Similar results could be frequently observed in other HRTEM images as shown in **Figure R1-3**, indicative of the abundant presence of dual interfaces over the $1\text{Ce}9\text{CoO}_x$ catalysts after the WGS reaction. Specifically, the metallic Co species typically found in the central zone, such as regions A and B illustrated in **Figure R1-4**, region A in **Figure R1-5**, and regions A and B in **Figure R1-6**, where the corresponding FFT patterns further proved the crystal information. More importantly, the co-presence of CoO and CeO_2 could always be identified in the same regions, such as region C, as illustrated in **Figure R1-4** and **R1-6**, where it is adjacent to the metallic Co species.

Figure R1-3. HRTEM images of spent $1\text{Ce}9\text{CoO}_x$ catalyst.

Figure R1-4. (a) HRTEM image for the selected area of $1\text{Ce}9\text{CoO}_x$ after transient WGS reaction, where the interfacial regions were marked with dotted lines; (b) and (c) the

corresponding FFT image obtained from the region A and B as illustrated in Figure R1-4a.

Figure R1-5. (a) HRTEM image for the spent $1\text{Ce}9\text{CoO}_x$ catalyst. (b) the corresponding FFT image obtained from region A as illustrated in Figure R1-5a.

Figure R1-6. (c) HRTEM image for the selected area of $1\text{Ce}9\text{CoO}_x$ after transient WGS reaction, where the interfacial regions were marked with dotted lines; (b-e) the corresponding FFT image obtained from region A, B, D and E as illustrated in **Figure R1-6c**.

In addition, the reasonability of the dual-interface structure was functionally proved. We supplementarily conducted experiments and characterizations upon four additional kinds of reference catalysts:

Catal. 1: the fresh $1\text{Ce}9\text{CoO}_x$ was directly evaluated as a catalyst without any reduction pretreatment (denoted by $\text{Co}_3\text{O}_4/\text{CeO}_2$);

Catal. 2: the $1\text{Ce}9\text{CoO}_x$ catalysts pretreated by $5\%\text{H}_2/\text{Ar}$ at $250\text{ }^\circ\text{C}$, for which the dominant phase is CoO and CeO_2 (denoted by CoO/CeO_2);

Catal. 3: the bare CeO_2 nanoparticles without cobalt species pretreated by $5\%\text{H}_2/\text{Ar}$ at $400\text{ }^\circ\text{C}$;

Catal. 4: the bare Co_3O_4 nanoparticles without CeO_2 species pretreated by $5\%\text{H}_2/\text{Ar}$ at $400\text{ }^\circ\text{C}$.

Firstly, the structure of reference catalysts was determined by XPS characterizations. One strong signal at 779.9 eV coupling with a weak satellite peak at 790.4 eV , as shown in $\text{Co } 2\text{p}$ XPS spectra for $\text{Co}_3\text{O}_4/\text{CeO}_2$, confirmed that Co_3O_4 was the initial phase (**Figure R1-7**). The binding energy of $\text{Co } 2\text{p}$ XPS for the CoO/CeO_2 catalyst was centered at 780.1 eV , which could be ascribed to the dominant presence of Co^{2+} (*Appl. Surf. Sci.* **2011**, *257*, 2717–2730). The characteristic distance of $\sim 6\text{ eV}$ between $\text{Co } 2\text{p}_{3/2}$ and the strong satellite peak indicates the dominant presence of CoO .

Figure R1-7. XPS spectra of three kinds of Co-based catalysts: Co_3O_4 , $\text{Co}_3\text{O}_4\text{-CeO}_2$ and CoO-CeO_2 . (a) $\text{Co } 2\text{p}$ XPS spectra and (b) $\text{C } 1\text{s}$ XPS spectra.

Figure R1-8. Comparison of the catalytic activity over 1Ce9CoO_x with other four reference catalysts. The CO conversion of the WGS reaction was controlled below 20% by modulating the space velocity of feed-gas (2%CO/10%H₂O/Ar).

Figure R1-9. *In-situ* Raman spectra were collected over CeO₂ (Catal. 3, a and c) catalyst and CoO-CeO₂ catalyst (Catal. 2, b and d) with sequentially switched CO and H₂O-containing atmosphere at 250 °C.

These catalysts displayed distinctly different catalytic performances in catalyzing WGS reactions (**Figure R1-8**), where the 1Ce9CoO_x catalyst pretreated at 400 °C vastly

outperformed other catalysts. Firstly, to determine the role of CoO-CeO₂ interface, we also conducted the *in-situ* Raman experiments over the bare CeO₂ (Catal. 3) and CoO/CeO₂ (Catal. 2) catalysts. Typical Raman active signals for CeO₂ centered at 463 cm⁻¹ (F_{2g} mode) and ca. 580 cm⁻¹ (D mode) could be identified during the CO-H₂O sequential switching tests (**Figure R1-9**). In the presence of CO molecules at 250 °C, the D mode signal could be enhanced over the CoO/CeO₂ catalyst, implying that the O species occupied vacancies could be easily removed in this step. However, the bare CeO₂ demonstrated negligible variation for the D mode at 580 cm⁻¹ as CO feeding inside, which proved the low activity of O species in the absence of the CoO-CeO₂ interface. More importantly, this consumption of O species upon the vacancies could be regenerated as H₂O molecules purge inside. This recyclability of O vacancies induced by the gas switching was indicative that the structural changes of CoO/CeO₂-CoO_{1-x}/CeO_{2-x} might participate in the CO oxidation step and H₂O dissociation step. The enhanced H₂O activation activity of the CoO/CeO₂ interface was also substantiated through kinetic tests compared with bare metallic Co catalysts. As shown in **Figure R1-10a**, the kinetic isotopic effect (KIE) derived from H₂O/D₂O switch for the CoO/CeO₂ catalysts was comparable to bare Co catalysts, indicating that the H₂O activation step was vital for both metallic Co and CoO/CeO₂ catalyst. In contrast, the H₂O apparent reaction order of CoO/CeO₂ is determined to be ca. 0.55 (**Figure R1-10b**), which was smaller than that of pure Co catalysts (0.89). The decreased H₂O reaction order indicated the higher coverage of H₂O-related species during the WGS reaction upon the surface of the CoO/CeO₂ catalyst, which was well in line with the enhanced H₂O dissociation ability as identified by *in-situ* Raman.

Figure R1-10. (a) The study of the kinetic isotope effect for the CoO-CeO₂ (Catal. 2) and Co₃O₄. (b) Apparent reaction order of H₂O and CO for the 1Ce9CoO_x and Co₃O₄ catalysts.

Figure R1-11. *In-situ* Raman spectra were collected over CeO₂ (Catal. 3, a and c) catalyst and CoO-CeO₂ catalyst (Catal. 2, b and d) with sequentially switched CO and H₂O-containing atmosphere at 250 °C.

Additionally, it should be noted that this reversible structural changes over the CoO/CeO₂ catalyst (Catal. 2) were different from that of the Co/CoO/CeO₂ catalyst. **Figure R1-11** shows a similar increase in the D mode signal when the CO feeds upon the surface of the 1Ce9CoO_x catalyst. Nevertheless, the 1Ce9CoO_x catalyst demonstrated a typical phonon vibration signal for Co₃O₄ as the H₂O molecules fed inside, while it was absent for the CoO/CeO₂. This result implied an oxidation behavior of the Co species proceeded upon the surface of the 1Ce9CoO_x catalyst, which should be strongly correlated with the presence of the Co/CoO interface. Videlicet, the H₂O molecules could dissociate on the CoO/CeO₂ interface, while the dissociated O-related species could not oxidize cobalt species without Co/CoO interface. As expected, this deduction was also supported by the CO-TPSR results for the CoO/CeO₂ catalyst. As shown in **Figure R1-12a**, the generated products follow a CO₂/H₂ ratio of ~2.3, implying that the O-related species involved in the surface reaction were hydroxy

groups rather than active O atoms. However, when the catalyst was pretreated at 5% H_2 /Ar by 400 °C, the 1Ce9CoO_x catalyst with as-formed Co/CoO/CeO₂ dual-interface demonstrated a distinctly different evolution during the CO-TPSR test ($CO_2/H_2 = \sim 75 \gg 2$), (**Figure R1-12b**), implying the dissociated O species from H₂O is active O atoms rather than –OH groups. Based on the above results, we could reasonably speculate that the CoO/CeO₂ interface is responsible for H₂O dissociation to OH groups, and the presence of Co/CoO could ulteriorly crack the OH group into active O atoms. The boosted H₂O dissociation ability thus enhances the WGS performance of the Co/CoO/CeO₂ dual interface.

Sorry for the tedious discussion above. In brief, the reasonability of the dual-interface within 1Ce9CoO_x catalysts could be proved in both the microstructure and functionality. To be more accurate, we have modified the description (*page 8, line 30; page 9, lines 1-12; page 13, lines 3-5; page 14, lines 1-21 and lines 26-30; page 15, lines 1-9*) and updated the corresponding figures as Fig. 2h, 2i, 4e, 4d and 4f; as well as Supplementary Fig. 20-23, Fig. 27 and Fig. 29 in the revised manuscript.

Figure R1-12. CO-TPSR profile for the (a) CoO-CeO₂ catalyst and (b) 1Ce9CoO_x

catalyst. The catalysts were pre-hydroxylated by 3% $\text{H}_2\text{O}/\text{Ar}$ at 250 °C before the experiments.

Comment 3. In this study, each in-situ characterization of prepared catalysts was conducted at different WGS reaction conditions (temperature and feed gas composition). The authors need to conduct the characterizations with the same condition of in-situ WGS reaction.

Response: Thanks for this valuable comment. Firstly, we are sorry for the mistake in the description of reaction temperature (300 °C) on *page 7, line 28*, which should be 250 °C as noted in the caption for Fig. 2 (Supplementary Fig. 17 in updated version). This careless oversight has been corrected, and we carefully checked the corresponding description in the manuscript. The conditions for the *in-situ* Raman and quasi *in-situ* XPS were paralleled, where the catalysts were conducted with the same atmosphere of 2% $\text{CO}/3\%\text{H}_2\text{O}/\text{Ar}$. As for the *in-situ* XRD experiments, the atmosphere containing only H_2/Ar was performed to monitor the precise phase transformation during pretreatment. However, it is problematic to collect the phase information under authentic WGS reaction conditions owing to the reaction chamber for *in-situ* XRD characterization is not water tolerant. Therefore, the initial components for the catalysts with various pretreatment could be identified according to the H_2 -TPR and *in-situ* XRD results. To exclude the composition effect for the (quasi) *in-situ* XPS and Raman experiments, we conducted additional catalytic tests with the same atmosphere of 2% $\text{CO}/3\%\text{H}_2\text{O}/\text{Ar}$. As shown in **Figure R1-11**, the 1Ce9CoO_x catalyst also demonstrate much better catalytic performance than the reference catalysts, suggesting that the distinction in the catalytic performance could be correlated to the difference observed by characterizations.

Thanks for the reviewer's reminder again. **A discussion on this point has been added in the revised manuscript (page 8, lines 4-7), and Figure R1-11 has been included in the updated manuscript as Supplementary Fig. 18.**

Figure R1-11. The comparison of the CO conversions for the WGS reaction performed at 260 °C over the 1Ce9CoO_x, Co₃O₄ and CeO₂ catalysts, where the atmospheres were comparable with the (quasi) *in-situ* Raman and XPS experiments (2%CO/3%H₂O/Ar).

Comment 4. The relevant paper should be cited in the reference.

Target-oriented water–gas shift reactions with customized reaction conditions and catalysts. Chemical Engineering Journal, 2023, 458, 141422

Response: Thanks for the reminder. **The relevant papers have been cited in the updated manuscript as ref. 28, 41, 42.**

To Reviewer #2

Fu et al. studied a complex Co-ceria system for the water gas shift reaction. The developed 1Ce9CoO_x catalyst shows high activity and exhibits different structure changes compared with reference catalysts. These results suggest that 1Ce9CoO_x possesses certain structural features that promotes the reaction.

Response: We thank the referee very much for the positive comment, which helped and guided us to further think about and polish our work considerably. We have tried our best to answer the questions from the reviewers. We hope that we have addressed all the questions satisfactorily.

***Comment 1.** To characterize the structure of catalysts, multiple techniques including XRD, TEM, XPS and Raman were used. However, I think this part needs to be improved. Currently, the set of tools reveals defects and possible Co species existing in the systems but there is no strong evidence connecting defects and different interfaces and how they participate into the reaction. Even for the interfaces, only based on TEM, it is hard to tell what interfaces exist in the system. Therefore, it is hard to claim that the improved activity is due to the synergistic effects of the proposed dual interfaces.*

Response: Thanks for the reviewer's suggestion. To further clarify the exact role of dual-interfaces in catalyzing WGS reaction over 1Ce9CoO_x catalysts, we conducted a series of experiments and characterizations upon additional four kinds of reference catalysts:

Catal. 1: the fresh 1Ce9CoO_x was directly evaluated as a catalyst without any reduction pretreatment (denoted by Co₃O₄/CeO₂);

Catal. 2: the 1Ce9CoO_x catalysts pretreated by 5%H₂/Ar at 250 °C, for which the dominant phase is CoO and CeO₂ (denoted by CoO/CeO₂);

Catal. 3: the bare CeO₂ nanoparticles without cobalt species pretreated by 5%H₂/Ar at 400 °C;

Catal. 4: the bare Co₃O₄ nanoparticles without CeO₂ species pretreated by 5%H₂/Ar at 400 °C.

As shown in Co 2p XPS spectra for Co₃O₄/CeO₂, the strong signal at 779.9 eV coupling with a weak satellite peak at 790.4 eV confirmed that the Co₃O₄ was the initial phase (**Figure R2-1**). The binding energy of Co 2p XPS for the CoO/CeO₂ catalyst was centered at 780.1 eV, which could be ascribed to the dominant presence of Co²⁺ (*Appl. Surf. Sci.* **2011**, 257, 2717–2730). The characteristic distance of 6.1 eV between Co 2p_{3/2} and the strong satellite peak indicates the dominant presence of CoO. The four kinds of catalysts displayed distinctly different catalytic performances in catalyzing WGS reactions (**Figure R2-2**), where the 1Ce9CoO_x catalyst pretreated at 400 °C vastly outperformed other catalysts. As remaindered by the reviewer, the authentic role of various interfaces was further investigated based on the four kinds mentioned above of Co-based catalysts with different interfacial structures.

Figure R2-1. XPS spectra of three kinds of Co-based catalysts: Co₃O₄, Co₃O₄-CeO₂ and CoO-CeO₂. (a) Co 2p XPS spectra and (b) C 1s XPS spectra.

Figure R2-2. Comparison of the catalytic activity over 1Ce9CoO_x with other four kinds of reference catalysts. The CO conversion of the WGS reaction was controlled below

20% by modulating the space velocity of feed-gas (2%CO/10%H₂O/Ar).

Figure R2-3. *In-situ* Raman spectra were collected over CeO₂ (Catal. 3, a and c) catalyst and CoO-CeO₂ catalyst (Catal. 2, b and d) with sequentially switched CO and H₂O-containing atmosphere at 250 °C.

Firstly, to dig out the role of the CoO-CeO₂ interface, we additionally conducted the *in-situ* Raman experiments over the bare CeO₂ (Catal. 3) and CoO/CeO₂ (Catal. 2) catalysts. Typical Raman active signals for CeO₂ centered at ca. 460 cm⁻¹ (F_{2g} mode) and 560 cm⁻¹ (D mode) could be identified during the CO-H₂O sequential switching tests (**Figure R2-3**). In the presence of CO molecules at 250 °C), the D mode signal could be enhanced over the CoO/CeO₂ catalyst, implying that the O species occupied vacancies could be easily removed in this step (**Figure R2-3d**). However, the bare CeO₂ catalyst demonstrated negligible variation for the D mode at 560 cm⁻¹ as CO feeding inside (**Figure R2-3c**), proving the low activity of O species in the absence of the CoO-CeO₂ interface. More importantly, the vacancies over the CoO/CeO₂ catalyst could be refilled as H₂O molecules purge inside, indicating that the active O-related species regenerated during this process. This recyclability of O vacancies, along with gas switching between reactant molecules, was indicative that the structural changes of

CoO/CeO₂-CoO_{1-x}/CeO_{2-x} might participate in the CO oxidation step and H₂O dissociation step over the CoO/CeO₂ catalyst. The enhanced H₂O activation activity of the CoO/CeO₂ interface was also substantiated through kinetic tests compared with bare metallic Co catalysts. As shown in **Figure R2-4a**, the kinetic isotopic effect (KIE) derived from H₂O/D₂O switch for the CoO/CeO₂ catalysts was comparable to bare Co catalysts (5.2 vs. 5.6), indicating that the H₂O activation step was vital for both metallic Co and CoO/CeO₂ catalyst. In contrast, the H₂O apparent reaction order of CoO/CeO₂ is determined to be ca. 0.55 (**Figure R2-4b**), which was smaller than that of pure Co catalysts (0.89). The decreased H₂O reaction order indicated the higher coverage of H₂O-related species during the WGS reaction upon the surface of the CoO/CeO₂ catalyst, which was well in line with the enhanced H₂O dissociation ability as identified by *in-situ* Raman experiments.

Figure R2-4. (a) The study of the kinetic isotope effect for the CoO-CeO₂ (Catal. 2) and Co₃O₄. (b) Apparent reaction order of H₂O and CO for the 1Ce9CoO_x and Co₃O₄ catalysts.

Additionally, it should be noted that this reversible structural changes over the CoO/CeO₂ catalyst were different from that of the 1Ce9CoO_x catalyst. As shown in **Figure R2-5**, a similar increase in the D mode signal was detected when the CO fed upon the surface of the 1Ce9CoO_x catalyst. Nevertheless, the 1Ce9CoO_x catalyst demonstrated a typical phonon vibration signal for the Co₃O₄ phase as the H₂O molecules fed inside, while it was absent for the CoO/CeO₂ catalyst (**Figure R2-4b**). This evolution implied an oxidation behavior of the Co species proceeded upon the 1Ce9CoO_x catalyst, which should be strongly correlated with the presence of the

Co/CoO interface. Videlicet, the H₂O molecules could dissociate on the CoO/CeO₂ interface, while the dissociated O-related species were unable to oxidize cobalt species without Co/CoO interface. As expected, this deduction was supported by the CO-TPSR results for the CoO-CeO₂ catalyst. Based on the prerequisite that adsorbed molecules reacted on the surface of the catalyst as $\text{CO} + \text{OH} \rightarrow \text{CO}_2 + 1/2\text{H}_2$, the CO-TPSR test typically can validate such an association process from the stoichiometric ratio (1/2) for produced H₂ and CO₂. As shown in **Figure R2-6a**, the generated products for the CoO-CeO₂ catalyst follow a CO₂/H₂ ratio of ~2.3, implying that the O-related species involved in the surface reaction should be hydroxy groups rather than active O atoms. However, when the Co/CoO interface was generated by pretreatment at 400 °C, the 1Ce9CoO_x catalyst with a dual-interface structure demonstrated a distinctly different CO₂/H₂ evolution during the CO-TPSR test (**Figure R2-6b**). Negligible H₂ was identified in the CO-TPSR profile over the hydroxylated 1Ce9CoO_x catalyst (CO₂/H₂ >> 2), indicating that the CO molecules might be oxidized by activated O atoms rather than OH groups upon the dual-interfaces. Based on the above results, we could reasonably speculate that the CoO/CeO₂ interface is responsible for H₂O dissociation to OH groups, and the presence of Co/CoO could ulteriorly crack the OH group into active O atoms. The boosted H₂O dissociation ability thus enhances the WGS performance of the Co/CoO/CeO₂ dual interface as shown in **Figure R2-2**.

Figure R2-5. *In-situ* Raman spectra were collected over CeO₂ (Catal. 3, a and c) catalyst and CoO-CeO₂ catalyst (Catal. 2, b and d) with sequentially switched CO and H₂O-containing atmosphere at 250 °C.

Figure R2-6. CO-TPSR profile for the (a) CoO-CeO₂ catalyst and (b) 1Ce9CoO_x catalyst. The catalysts were pre-hydroxylated by the 3%H₂O/Ar at 250 °C before experiments.

Thanks again for the reviewer's valuable comment. **In the revised manuscript, we added further discussion on page 13, lines 3-5; page 14, lines 1-21 and lines 26-30; page 15, lines 1-9. The corresponding figures have been supplemented in the updated version as Fig. 4d-4f, Supplementary Fig. 27 and Fig. 29.**

Comment 2. XRD has limitations in detecting species with small sizes or with disordered structures. Need to provide additional evidence to show that in the fresh 1Ce9CoO_x, Co is in only in Co₃O₄ phase.

Response: Thanks for this comment. We agree with the reviewer that XRD is limited to identifying the amorphous species and further supplemented the XPS and Raman

characterizations for the fresh $1\text{Ce}9\text{CoO}_x$. As shown in **Figure R2-7**, three active Raman modes typical of the spinel structure were identified at 193, 473 and 680 cm^{-1} , which were assigned to the F_{2g} , E_g and A_{1g} photo modes of Co_3O_4 , respectively. For the Co 2p XPS spectrum (**Figure R2-8**), there is one strong signal at 779.9 eV coupling with a weak satellite peak at 790.4 eV ($\Delta = \text{ca. } 10.5\text{ eV}$) as shown in Co 2p XPS spectra, which is a typical feature for $\text{Co}^{2+}/\text{Co}^{3+}$ in the Co_3O_4 spinel phase. These results above further confirmed that Co_3O_4 was the initial phase for the fresh $1\text{Ce}9\text{CoO}_x$ catalysts.

Thanks for the reviewer's reminder again. **A discussion on this point has been added in the test (page 7, lines 4-9), and Figure R2-7 and R2-8 have been included in the updated manuscript as Supplementary Fig. 3b and Fig 2d-2f, respectively.**

Figure R2-7. Raman spectrum of fresh $1\text{Ce}9\text{CoO}_x$ catalyst.

Figure R2-8. (a) Co 1p, (b) C 1s, (c) O 1s, and (d) Ce 3d XPS spectra of fresh 1Ce9CoO_x catalyst.

Comment 3. It is hard to tell the size of CoO_x and CeO_x species by using STEM. In addition, STEM is a local technique. Suggest to performing XAS measurements to provide complementary results.

Table R1. EXAFS fitting results (*R*: distance; CN: coordination number) of the used catalysts.

Sample	Co-O		Co-Co		σ^2 (Å ²)	ΔE_0 eV
	R (Å)	CN	R (Å)	CN		
1Ce9CoO _x -1	1.93±0.01	5.0±0.5	2.85±0.02	3.4±0.9	0.006 (O)	-
			3.35±0.02	4.8±1.4	0.009 (Co)	2.6±1.2
			2.50±0.01	6.9±0.4	0.006 (O)	-
1Ce9CoO _x -2	1.90±0.01	3.4±0.2	2.81±0.01	4.3±0.6	0.009 (Co)	2.2±1.7
			3.37±0.02	3.6±0.7		
Co foil	-	-	2.484±0.003	12	0.007 (Ru)	7.3±0.5

Figure R2-9. (a) XANES and (b) EXAFS results of spent 1Ce9CoO_x catalysts for the first time, where the catalyst was exposed to air for more than one week without any protection.

Figure R2-10. (a) XANES and (b) EXAFS results of spent $1\text{Ce}9\text{CoO}_x$ catalysts collected for the second time, where the catalyst was protected by N_2 before experiments.

Response: Thanks for the reviewer's suggestion. We agreed that the limited numbers of STEM images in the manuscript could not offer reliable average size information of $1\text{Ce}9\text{CoO}_x$ catalysts. Following the reviewer's advice, we further conducted an *ex-situ* XAFS test over the spent $1\text{Ce}9\text{CoO}_x$ catalyst. As shown in **Figure R2-9**, the XAFS results illustrate that the Co species for spent $1\text{Ce}9\text{CoO}_x$ catalyst only comprise Co_3O_4 and CoO while the metallic Co was absent. However, this finding was at variance with XRD and XPS results, in which the presence of metallic Co was undoubtedly confirmed. The spent $1\text{Ce}9\text{CoO}_x$ catalyst for XAFS experiments was carefully collected after the temperature of the catalyst bed cooled to room temperature. Nevertheless, owing to the instability of the metallic Co and CoO , the uncertain waiting time (as long as 10 days in the final) for the XAFS experiment might result in the inevitable oxidation by the air for the spent $1\text{Ce}9\text{CoO}_x$ catalyst. For this regard, we took another XAFS experiment (**Figure R2-10**) where the spent $1\text{Ce}9\text{CoO}_x$ catalyst was sealed in a bottle full of N_2 . As expected, the presence of metallic Co species and CoO were confirmed over the $1\text{Ce}9\text{CoO}_{x-2}$ catalyst, in which the content of Co is ca. 27%, and the fraction of CoO is 21% as determined by the XANES fitting results (**Figure R2-11**). The abundant Co_3O_4 might be stemmed from the unavoidable oxidation of the metastable Co species or generated from the redox reaction cycles as illustrated by *in-situ* Raman results (**Figure R2-5**). The coordinated number (CN) of the Co-Co bond ($d = 2.50 \pm 0.01$) is 6.9 ± 0.4 , which corresponded to Co nanoparticles with an average diameter lower than 2 nm (*J.*

Am. Chem. Soc. **2006**, *128*, 3956–3964). However, the crystallized metallic Co species derived from XRD patterns (**Figure R2-12**) based on the Scherrer Formula was 8.4 nm (Co), which was inconsistent with coordination numbers as determined by the XAFS results. Two inductions might induce this contradiction: the oxidation process from the metallic Co to Co₃O₄ during XAFS experiments; the XRD pattern was limited to determine the crystallized size below 2 nm.

To precisely illustrate the size of CoO_x and CeO_x species, two additional ways were used to obtain crystal diameters based on HRTEM and XRD results. The boundary of the CeO₂ nanoparticles could be identified according to the HRTEM images, where the lattice distance of CeO₂ ($d_{(111)} = 0.31$ nm; $d_{(200)} = 0.27$ nm) could be determined as shown in **Figure R2-13a**. The average diameter of CeO₂ species derived from statistical results based on more than 100 nanoparticles was about 4.0 nm (**Figure R2-13b**), in which the sizes of the nanoparticles were counted after confirming phase information. Unfortunately, only the nanoparticles located on the edge region of assembled structure could be clearly distinguished based on the TEM images. As shown in **Figure R2-13**, the crystallized CeO₂ species were rich in the outermost layer, which resulted in difficulty in getting reliable information on Co-related species based on the statistical results. As shown in the HRTEM images, merely ~30 nanoparticles with distinct boundaries could be identified for CoO species (**Figure R2-14**), where the average diameter, for reference only, was calculated to be ~4.3 nm. The mean size derived from the XRD results based on the Scherrer Formula is ca. 4.3 nm and 4.2 nm for the CeO_x and CoO nanoparticles, respectively, in line with the statistical results derived from TEM images.

Thanks again for the helpful suggestion. The updated manuscript has supplemented the corresponding contents (page 5, lines 10-12, lines 17-18, lines 28-29). In addition, the manuscript in the updated version has included the corresponding figures and table as Supplementary Fig. 11, Fig. 12, Fig. 15, Fig.16 and Table 3, respectively.

Figure R2-11. XANES fitting result for the 1Ce9CoO_x used catalyst.

Figure R2-12. XRD pattern of 1Ce9CoO_x collected after WGS reaction or *in-situ* reduction by 5% H_2 /Ar 400 °C.

Figure R2-13. Statistic distribution of CeO₂ particle size.

Figure R2-14. Statistic distribution of CoO particle size.

Comment 4. Could you please compare the Ce and Co XPS spectra collected before and after the reaction? Now Fig. 2d,e only show the spectra collected after the reaction. Without comparison, one cannot tell how Co and Ce species change due to the reaction.

Response: Thanks for the kind reminder. According to the reviewer's suggestion, we supplementally conducted an XPS experiment over the fresh 1Ce9CoO_x catalyst. As shown in **Figure R2-15**, the Co 2p, Ce 3d and O 1s *ex-situ* XPS spectra were carefully deconvoluted and compared. The fresh 1Ce9CoO_x catalyst was characterized by the Co 2p_{3/2} binding energy at 779.9 eV and shake-up satellite signal at 790.4 eV with a low intensity, which was the typical feature for the Co²⁺/Co³⁺ ions in the Co₃O₄ spinel structure. After the light-off reaction test (2%CO/10%H₂O/Ar), the binding energy of Co species observed at 780.1 eV coupling with a strong satellite peak at 786.2 eV higher was typically ascribed to the CoO phase. The spin-orbit coupling peak at 15.5 eV was also a characteristic feature of CoO, which could be clearly identified for the spent 1Ce9CoO_x catalyst. In addition, a small shoulder peak at 778.0 eV ascribed to metallic Co was detected after the WGS reaction. Based on the above comparison, an unambiguous change from Co₃O₄ to Co/CoO were induced by the pretreatment and the WGS reaction for the 1Ce9CoO_x catalyst. The Ce 3d spectrum was deconvoluted and labeled according to Burroughs formalism (**Figure R2-15b and R15d**), where the fitted u' and u⁰ resulted from Ce³⁺. Interestingly, compared with fresh catalyst, the Ce³⁺ fraction is significantly boosted from 0.15 to 0.28, indicating that the amount of O_v-Ce increased after the WGS reaction. A similar conclusion could be derived from the O 1s

results, in which the vacancy-related O species (O1 and O2) were increased from 0.40 to 0.68 after the WGS reaction.

Thanks for the valuable suggestion. **We revised the manuscript on page 7, lines 4-11 and 13-22. The corresponding figures for XPS spectra collected before and after the WGS reaction have been included in the updated version as Fig. 2d-2f.**

Figure R2-15. XPS results of 1Ce9CoO_x-fresh and 1Ce9CoO_x-used catalysts. (a and d) Co 2p, (b and e) Ce 3d, (c and f) O 1s XPS spectra.

Comment 6. From Fig.2d, it is hard to tell the existence of Co metallic. Could you please do the fitting to confirm its existence? Fig. S12b also plotted the Co 2p XPS of used 1Ce9CoO_x but the spectral features are different from those in Fig. 2d. Please examine.

Response: Thanks for this valuable suggestion. Firstly, we are sorry for the careless mistake during calibrating the XPS spectra, where the calibration position of C1s for Figure S12 was 284.0 eV, and it was 284.8 eV for Fig. 2d. It has been carefully recorrected in the updated version and thanks for the reviewer's reminder.

We have fitted the Co 2p XPS spectra, and the deconvoluted results are illustrated in

Figure R2-16. Three peaks were fitted to the Co(II) 2p XPS spectrum according to the previous report (*Appl. Surf. Sci.* **2011**, 257, 2717–2730). The deconvoluted peak fixed at 780.1 eV was the characteristic peak for Co²⁺ according to previous reports, where a strong shake-up satellite signal at ca. 5.5 eV higher B.E position and a spin-orbit coupling peak at ca. 15.5 eV higher B.E position could be determined for the Co²⁺ identification. The robust shake-up satellite indicates the presence of paramagnetic Co²⁺ species on the surface of 1Ce9CoO_x catalyst after *in-situ* pretreatment; this signal is absent in diamagnetic Co³⁺ species. The presence of metallic Co⁰ was proven by the fitting results, in which a small yellow deconvoluted peak centered at 778.1 eV was preserved after the mathematical fit treatment (~3.5%).

Figure R2-16. Quasi *in-situ* XPS results of 1Ce9CoO_x: (a) Co 2p (b) Ce 3d and (c) O 1s XPS spectra. The XPS spectra were collected after *in-situ* pretreated by 5%H₂/Ar at 400 °C for 1 h and *in-situ* WGS reaction at 250 °C for 1 h (2%CO/~3%H₂O/Ar).

It should be noted that the metallic Co species could be identified over the XRD patterns collected after the light-off WGS reaction test (**Figure R2-12**). In addition, the apparent presence of Co⁰ is identified over the 1Ce9CoO_x catalysts by *ex-situ* XAFS (~27%, **Figure R2-11**) and *ex-situ* XPS spectra (~6.2%, **Figure R2-15**) collected after the light-off WGS reaction test. However, the content of metallic Co is extremely low as determined by quasi *in-situ* XPS results (~3.5%, **Figure R2-16**). In addition, the O_v amount derived from the O 1s or Ce 3d spectrum collected after quasi *in-situ* experiments ((O₂+O₃)/O_{total}) = 0.34; Ce³⁺/Ce_{total} = 0.24) was lower than that after *ex-situ* light-off WGS reaction ((O₂+O₃)/O_{total}) = 0.68; Ce³⁺/Ce_{total} = 0.28). The above experimental results were somewhat 'self-contradictory' and interested us in figuring

out the authentic origin of this phenomenon.

The difference in the spectral features in Fig. 2d and Fig. S12b is primarily induced by the distinct pretreated conditions prior to the XPS experiments. As for the *ex-situ* Co 2p XPS spectra illustrated in Fig. S12b, the spectrum was collected over spent 1Ce9CoO_x catalyst after the light-off WGS reaction (2%CO/10%H₂O/Ar), where the maximum temperature was 320 °C. In contrast, to be paralleled with other *in-situ* experiments conducted in this work, the pretreated temperature for the quasi *in-situ* XPS experiment was 250 °C. Additionally, limited by the instrument requirement, relatively low partial pressure (~0.3 bar, ~3%) of H₂O coupling with 2%CO was fed inside the pretreated chamber before collecting spectra, which generated a relatively milder reducing atmosphere as compared with the pretreated condition for *ex-situ* XPS experiments (Fig. S12 in the supplementary). Therefore, the differentiated temperatures and H₂ concentration in the atmosphere for the *ex/in-situ* experiments consequently resulted in a distinct portion of Co⁰ species as detected by XPS characterizations.

Even though this difference is somewhat reasonable in our opinion, while inspired by the reviewer's rigorous advice and valuable reminder, we reconsider the plausible origin of the structural information derived from multiple characterizations as discussed in the manuscript. The different features in Fig. 2d and Fig. S12b indicate the extreme sensitivity of the phase evolution on the reaction condition. The structural dependence on reaction conditions, especially surface structure determined by XPS characterization, is consistent with the redox mechanism proposed in the manuscript. As discussed in response to comment 1, the *in-situ* Raman spectra (**Figure R2-3** and **R5**) proved that the facile and efficient proceeding redox cycles of Co species during sequential CO-H₂O switches were beneficial from the Co/CoO/CeO₂ dual interface. Videlicet, the activation process of reactant molecules (CO and H₂O) results in the dynamic structures of the catalyst surface along with changing the partial pressure of feed-gas or reaction temperatures.

In brief, based on the *ex-situ* characterizations, the tremendous structural change could be sufficiently illustrated for the 1Ce9CoO_x catalyst collected before and after the

WGS reaction, as illustrated in **Figure R2-15**, which strongly indicates the structural evolution from $\text{Co}_3\text{O}_4\text{-CeO}_2$ to Co/CoO/CeO_x . However, because the metallic Co or CoO species were susceptible to oxygen, the results of *ex-situ* characterizations for spent $1\text{Ce}9\text{CoO}_x$ catalysts were susceptible to interference from plausible oxidation (**Figure R2-9 and R10**). For this reason, we conducted quasi *in-situ* XPS experiments on the $1\text{Ce}9\text{CoO}_x$ to exclude interference from the possible oxidation. Despite that the quasi *in-situ* results were somewhat unmatched by *ex-situ* results owing to the redox reaction mechanism, the dominant presence of Co^{2+} species derived from quasi *in-situ* experiment condition proves the CoO was indeed maintained during the WGS reaction over the $1\text{Ce}9\text{CoO}_x$ catalyst instead of being oxidized by air.

Sorry for the incomplete description of the spent catalyst in the caption for Fig. 2, which was carefully optimized in the updated version. In addition, we appreciate the rigorous comment from the reviewer, which reminded us to analyze the ignorable difference in XPS results. We reconsider our XPS results and reorganize them in the updated manuscript. **The corresponding discussion has been included in the updated version on page 10, lines 23-30, and the figures have been updated as Fig. 2d-2f and Supplementary Fig. 17.**

Comment 8. In XPS, the observed Co^{2+} is assigned to Co^{2+} in CoO or CoO_{1-x} . Is it possible that Co^{2+} is from the interface of Co/CeO₂?

Response: We agree with the reviewer that the Co^{2+} species stems from the interface structure of Co/CeO₂ catalysts. In this work, we highlight the importance of interfacial structure within Co/CeO₂ catalysts and develop an alternative strategy to fabricate stable dual active interfaces resorting to the boosted O spillover from CeO_{2-x} islets. Owing to the electronic interaction between metal (denoted by M) and oxides support (denoted by M'O_x), the electron transfer between interfacial metal atoms and oxides resulted in partially positive/negative metal sites, which have been extensively investigated previously. However, the electronic interaction typically affects these metal atoms directly coordinated with O atoms (M-O-M'), indicating that only limited

numbers of M atoms located at the perimeter sites would be changed to $M^{n+/n-}$. The fraction of perimeter sites in a particle with a particular geometry diameter could be roughly estimated (*J. Am. Chem. Soc.* **2012**, *134*, 4700–4708). The expected value calculated based on the Co particle size (d) should be ca. ~ 0.01 (~ 8 nm, XRD results) or ~ 0.1 (~ 2 nm, XAFS results), where the fraction of perimeter site (N) is calculated based on the following equation.

$$N = 0.46d^{1.8}$$

The as-calculated interfacial Co based on XAFS is well in line with the H₂-TPR results (**Figure R2-17a**), in which the fraction of the γ peak (assigned to interfacial Co species) is ca. 0.15. As determined by the *ex-situ* XANES results (**Figure R2-11**), the fraction of CoO species is about 27% over the spent 1Ce9CoO_x catalyst, where the catalysts were collected after the WGS reaction at 250 °C with an atmosphere of 2%CO/10%H₂O/N₂. In addition, the abundant presence of CoO species ($> 90\%$) was further confirmed by XPS results (**Figure R2-15** and **R2-16**). These results above indicate that the Co²⁺ species were not merely derived from the interfacial Co atoms directly coordinated to O atoms from CeO₂ but also the crystalized CoO_x species. As proven by the *in-situ* XRD (**Figure R2-17c**), in the absence of CeO₂, the Co₃O₄ was directly reduced to metallic Co at around 300 °C. In contrast, the local interfacial confinement between Co and CeO₂ helps to maintain a metastable CoO state in a wide temperature range of 300-500 °C. Interestingly, the role of support oxides in stabilizing metastable CoO phase was recently also found by Bert M. Weckhuysen's group and Xin-He Bao's groups, in which CoO_x species were maintained by forming an interface with TiO_x or ZnO (*Nat. Commun.* **2022**, *13*, 324.; *J. Am. Chem. Soc.* **2023** *145*, 17056-17065).

The electron energy-loss spectrum (EELS) is sensitive to the chemical environment of the selected region, which could thus be used to probe the oxidation state. As shown in **Figure R2-18**, the Ce species were generally distributed among the space interval of Co species with a diameter in the range of 3–5 nm, which was consistent with the discussion in response to comment 3. The gradient evolution of Co and Ce intensity

(Figure R2-18b) was reversed along the selected dotted line in Figure R2-18a, indicating the presence Co-CeO₂ interface within in probed region. In addition, we collected the Co-L and Ce-M spectra within two typical regions: Ce-rich region (R-A) and Co-rich region (R-B), and. The creation of Co oxide was proved by the energy shift within the Co-L_{2,3} edge spectra when the Co atoms were overlapped with the CeO₂ species (*Adv. Mater.* **2019**, *31*, 1900062; *Adv. Funct. Mater.* **2021**, *31*, 2101239). The intensity ratio of L₃/L₂ is determined by the occupation state of 3d-states, which thus could monitor the electronic change induced by the creation of oxygen vacancies within the interface region. The L₃/L₂ ratio for Co L-edge collected in R-A was ca. 4.8, while the corresponding values calculated for R-B decreased to 3.2. The increased L₃/L₂ ratio with the dominant presence of CeO₂ might indicate that the lower valence state of Co species is induced by the creation of oxygen vacancies within the CoO-CeO₂ interface.

In general, two inductions should be contributed to the formation of Co²⁺ species as determined by XPS results: Co atoms were directly connected with CeO₂ via Co-O-Ce; the metastable CoO_x species were maintained due to the Co-CeO₂ interfacial confinement. Thanks for the reviewer's valuable comment. **We have further discussed the origin of the Co²⁺ in the revised manuscript on page 8, lines 21-26. Figure R2-18 has been included in the updated version as Supplementary Fig. 20.**

Figure R2-17. (a) H₂-TPR profile of 1Ce9CoO_x, Co₃O₄, and pristine CeO₂ samples. The *in-situ* XRD patterns were collected under 5%H₂/Ar atmosphere with different temperatures for the (b) Co₃O₄ and (c) 1Ce9CoO_x catalysts.

Figure R2-18. (a) STEM-EELS mapping results, (b) EELS intensity profiles of Co-L, Ce-M, and O-K edges following the selected dotted line in Figure R2-18a and (c) Co-L and Ce-M edge EELS spectra collected within various regions over the spent $1\text{Ce}_9\text{CoO}_x$ catalyst.

Comment 9. I don't think one can use the atomic ratio of O/Co to claim that Co is in CoO_{1-x} . XPS detected not only Co^{2+} but also Co^0 . The ratio of O/Co must be smaller than 1. Besides, if Co is CoO_{1-x} , the oxidation state of Co would be different from Co^{2+} , will you see a peak shift in XPS?

Response: Thanks for the reviewer's professional advice. I agree with the reviewer about the distraction from metallic Co species. As discussed above, the Co 2p spectra were fitted following the reviewer's suggestion. As shown in **Figure R2-16**, the fraction of metallic Co species is ca. 3.5%, and the ratio of O/Co is re-calculated to be ~ 0.71 . However, as the reviewer pointed out, it is less rigorous to claim the presence of O_v by comparing the experimental O/Co ratio to the stoichiometric ratio, which was rarely reported in previous findings. Owing to the complexity of the convolution for the Co 2p XPS signal (*Surf Interface Anal.* **2021**, *53*, 475–481; *ACS Catal.* **2018**, *8*, 9625–9636), it is extremely difficult to identify a reliable peak shift induced by oxygen

vacancies for the Co^{2+} species.

To make reliable speculation about the O_v structures based on the XPS results, we supplemented the additional investigation on various reference catalysts, including fresh $1\text{Ce}9\text{CoO}_x$ (**Figure R2-15a-R2-15c**), spent $1\text{Ce}9\text{CoO}_x$ (after light-off WGS reaction, **Figure R2-15d-R2-15f**), $1\text{Ce}9\text{CoO}_x\text{-}250\text{H}$ (after pretreatment by 5% H_2/Ar at 250 °C, **Figure R2-19**) and bare CeO_2 (**Figure R2-20**). The O 1s XPS spectra were deconvoluted into two subpeaks (O1, O2 and O3). Based on previous reports, the O1 signal centered at ca. 529.7 eV could be ascribed to lattice O arising from Co-O or Ce-O bond. The O2 and O3 peak at ca. 531.5 eV and 532.2 eV corresponds to O^{2-}/O^- or OH species located on the vacancy sites (*Adv. Funct. Mater.* **2021**, *31*, 2101239). The O 1s XPS spectra of fresh $1\text{Ce}9\text{CoO}_x$ and CoO-CeO_2 catalyst demonstrate a value of ~ 0.40 and 0.38 for the O2 and O3 species according to the deconvoluted results. In comparison, the fraction of O2 and O3 for the bare CeO_2 was only ~ 0.17 , indicating that the Co-CeO_2 interface could augment the number of defective sites upon the catalyst surface. The significantly boosted number of vacancies over $1\text{Ce}9\text{CoO}_x$ was also confirmed by the O 1s spectra collected after quasi *in-situ* reaction at 250 °C (2% $\text{CO}/3\%\text{H}_2\text{O}/\text{Ar}$). As shown in **Figure R2-16**, the fraction of oxygen vacancies derived from O 1s XPS results was about ~ 0.34 .

When the spent $1\text{Ce}9\text{CoO}_x$ catalyst was collected after the WGS reaction at 320 °C (2% $\text{CO}/10\%\text{H}_2\text{O}/\text{Ar}$), the O 1s XPS spectra illustrated a much higher ratio of vacancy-related O species (0.68). In addition, the $\text{Ce}^{3+}/(\text{Ce}^{3+}+\text{Ce}^{4+})$ is ~ 0.28 for the spent catalysts, and the total $\text{CeO}_x/(\text{CeO}_x+\text{CoO}_x)$ species upon the surface layer as determined by XPS results was ~ 0.12 , suggesting that the O species related with $\text{O}_v\text{-Ce}^{3+}$ structure should be a minor component (less than 0.03). However, it was found that the amount of O_v (0.68) determined by the O 1s XPS results was significantly higher than the atomic amount of Ce^{3+} (< 0.03). Therefore, this evidenced that the defective-related species over the $1\text{Ce}9\text{CoO}_x$ catalyst were predominantly stemmed from the defective structure of CoO_x , which was consistent with the EELS result as shown in **Figure R2-18**.

Thanks for the reviewer's very professional advice. We have deleted the former discussion about the atomic ratio of O/Co and further modified the corresponding discussion about Co 2p XPS results on *page 7, lines 22-25*. Figures R19 and R20 were included in the updated version as Supplementary Fig. 28 and Fig. 29.

Figure R2-19. XPS results of CoO-CeO₂: (a) Co 2p, (b) Ce 3d and (c) O 1s XPS spectra.

Figure R2-20. XPS results of bare CeO₂: (a) Ce 3d and (b) O 1s XPS spectra.

Comment 10. Page 5, "The amount of the CeO₂-Co interfacial sites within the 1Ce9CoO_x catalyst was defined by the perimeter outline of small CeO_{2-x} particles with favored thermostability, which consequently resist the loss of active interfacial Co-CeO₂ sites under reaction condition." Page 7, "The differentiated O/Co ratios in the surface region of catalysts acquired over quasi in-situ experiments clearly showed that the active phase is nonstoichiometric cobalt monoxide (CoO_{1-x}) rather than CoO, ..." How do you know these species or sites are active? Because they exist in the catalyst?

Response: Thanks for pointing out the inappropriate description and insufficient discussion about the active sites in the manuscript. We agree with the reviewer that the contribution of active sites has not been confirmed yet based on the results on page 5

or 7, and deduction about active sites should not be prematurely drawn. Therefore, the description of *"The differentiated O/Co ratios in the surface region of catalysts acquired over quasi in-situ experiments clearly showed that the active phase is nonstoichiometric cobalt monoxide (CoO_{1-x}) rather than CoO , ..."* on page 7 **has been deleted in the updated version**; the description about *"The amount of the CeO_2 -Co interfacial sites within the $1\text{Ce}9\text{CoO}_x$ catalyst was defined by the perimeter outline of small CeO_{2-x} particles with favored thermostability, which consequently resist the loss of active interfacial Co- CeO_2 sites under reaction condition."* on page 5 was changed as *"The amount of the CeO_2 -Co interfacial sites within the $1\text{Ce}9\text{CoO}_x$ catalyst was defined by the perimeter outline of small CeO_{2-x} particles with favored thermostability, which consequently resist the loss of interfacial Co- CeO_2 sites under reaction condition."*

Comment 11. What is the evidence for the formation of Co/ CoO_{1-x} interface?

Response: Thanks for the comment. To validate the presence of the $\text{CoO}_{1-x}/\text{Co}$ interface as proposed in the manuscript, we conducted experiments to clarify it from two aspects: confirming the presence of $\text{CeO}_{2-x}/\text{CoO}_{1-x}$ and $\text{CoO}_{1-x}/\text{Co}$ interface through the characterization results of spent catalysts; illustrating the functional cooperativity of each interface through comprehensive comparison over various reference catalysts. Firstly, we further supplement microscopy evidence for the presence of $\text{CeO}_{2-x}/\text{CoO}_{1-x}$ and $\text{Co}/\text{CoO}_{1-x}$ interface within the spent catalysts. As shown in **Figure R2-21**, the crystal lattice ascribed to $\text{CeO}_2(100)$, $\text{CoO}(111)$, and $\text{Co}(100)$ could be clearly identified, where the edged region, as marked by the dotted line, was correspondingly contributed to $\text{CoO}_{1-x}/\text{Co}$ or $\text{CeO}_{2-x}/\text{CoO}_{1-x}$ interfaces. Similar results could be frequently observed in other HRTEM images as shown in **Figure R2-22-R2-25**, indicative of the abundant presence of dual interfaces over the $1\text{Ce}9\text{CoO}_x$ catalysts after the WGS reaction. Specifically, the metallic Co species generally found in the central zone, such as regions A and B illustrated in **Figure R2-23**, region A in **Figure R2-24**, regions A and B in **Figure R2-25** and region A in **Figure R2-26**, where the corresponding FFT patterns

further proved the crystal information. More importantly, the co-presence of CoO and CeO₂ could always be identified in the same regions, such as region C in **Figure R2-23** and **R25**, where it is adjacent to the metallic Co species.

To further prove these findings based on local areas, additional evidence proving the presence of interaction acquired after *in-situ* treatment would be advantageous. The reduction behavior of Co-based species determined by the H₂-TPR test could strongly hint at the phase information and chemical interaction between heterogeneous species (*Science* **2023**, 380, 1174–1178). Therefore, we collected the H₂-TPR profile for the 1Ce9CoO_x catalysts after the *in-situ* WGS reaction (250 °C, 2%CO/3%H₂O/Ar), which can rule out the interference from oxidation during other *ex-situ* characterizations. As shown in **Figure R2-27**, the first round of H₂-TPR results demonstrated a typical evolution for the multi-step reduction of Co₃O₄ species (*Nat. Catal.* **2020**, 3, 526–533.). After *in-situ* WGS reaction treatment, the reduction peak at 400–600 °C implied that the interfacial CoO species was maintained during the WGS reaction. Interestingly, another sharp peak centered at 172 °C was observed when the 1Ce9CoO_x was exposed to the feed-gas, which might belong to the reduction of active O species on the surface. On the one hand, these H₂-TPR results evidenced the presence of the CoO_x species even after H₂-pretreatment and subsequent WGS reaction; on the other hand, the dominant presence of active O species was in line with the proposed redox mechanism upon the surface of the catalyst.

Figure R2-21. (a-d) Atomic-resolution HAADF-STEM images for the selected area of $1\text{Ce}9\text{CoO}_x$ after transient WGS reaction to determine the microstructure of spent catalysts. The CoO nanoparticles are surrounded by a few small CeO_2 nanoparticles with a diameter of about 2–5 nm.

Figure R2-22. HRTEM images of spent $1\text{Ce}9\text{CoO}_x$ catalyst, where the interfacial regions were marked with dotted line.

Figure R2-23. (a) HRTEM image for the selected area of $1\text{Ce}9\text{CoO}_x$ after transient WGS reaction, where the interfacial regions were marked with a dotted line; (b) and (c) the corresponding FFT image obtained from the region A and B as illustrated in Figure R2-23a.

Figure R2-24. (a) HRTEM image for the spent $1\text{Ce}9\text{CoO}_x$ catalyst. (b) the corresponding FFT image obtained from region A as illustrated in Figure R2-24a.

Figure R2-25. (c) HRTEM image for the selected area of $1\text{Ce}9\text{CoO}_x$ after transient WGS reaction, where the interfacial regions were marked with a dotted line; (b-e) the corresponding FFT image obtained from the region A, B, D and E as illustrated in Figure R2-25c.

Figure R2-26. (a) HRTEM image for the spent $1\text{Ce}9\text{CoO}_x$ catalyst. (b) the corresponding FFT image obtained from the region A as illustrated in Figure R2-26a.

Figure R2-27. H₂-TPR profiles for the fresh $1\text{Ce}9\text{CoO}_x$ (red) and spent $1\text{Ce}9\text{CoO}_x$ pretreated by *in-situ* WGS reaction (blue).

Secondly, the presence of the $\text{CoO}_{1-x}/\text{Co}$ interface could be functionally proved by comparing it with other reference catalysts, which have been detailed discussed in response to comment 1. When the metallic Co is absent, such as Catal-1 ($\text{CoO}-\text{CeO}_2$) and Catal-2 ($\text{Co}_3\text{O}_4-\text{CeO}_2$), the performance of catalysts was much inferior as compared with the $1\text{Ce}9\text{CoO}_x$ catalyst (**Figure R2-2**), implying the critical role of Co phase in proceeding efficient reaction cycles (**Figure R2-3** and **R2-5**). To eliminate the crucial role of CoO_x for catalyzing the WGS reaction within the $1\text{Ce}9\text{CoO}_x$ catalyst, the $1\text{Ce}9\text{CoO}_x\text{-PM}$ were chosen as reference catalysts, where the $1\text{Ce}9\text{CoO}_x\text{-PM}$ were prepared by physically mixing nine atomic units of Co nanoparticles and one atomic unit of CeO_2 nanoparticles. The $1\text{Ce}9\text{CoO}_x\text{-PM}$ demonstrated a much lower CO conversion rate as compared with the $1\text{Ce}9\text{CoO}_x$ catalyst (7.2 vs. 169.3 $\text{mmol}_{\text{CO}}/\text{g}_{\text{cat}}/\text{h}$), suggesting the inferiority of independent Co and CeO_2 phase in catalyzing WGS reaction. As shown in **Figure R2-28**, even though both Co species and CeO_2 species were present in this catalytic system, there were no detectable signals of O-vacancies (ca. 550 cm^{-1}) identified over the *in-situ* Raman spectra for $\text{Co}/\text{CeO}_2\text{-PM}$ catalyst as co-feeding CO and H_2O molecules inside. These results evidenced that the presence of CoO_x induced by chemical bonding with CeO_2 species is essential for the reversible

consumption and regeneration process of O-vacancies. As shown in **Figure R2-29**, the H₂O and CO apparent reaction order was ca. 0.9 and -1.2, respectively, implying analogical sorption behavior compared with bare metallic Co catalyst. Evidence for the synergistic effect can also be observed over the *in-situ* Raman results. In brief, the above results suggest that the co-presence of Co and CoO_x species were functionally requisite for the boosted WGS catalytic performance over the 1Ce9CoO_x catalyst.

Thanks for the valuable comment from the reviewer. **We have strengthened the proof for the presence of dual interface in the revised manuscript, and the corresponding discussion has been included on page 8, line 30; page 9, lines 1-9; page 14, lines 17-21; page 15, lines 7-9.** Figure R2-26 has been supplemented in the manuscript as Fig. 2h and 2i, and the other corresponding figures were included as Supplementary Fig. 21-23 and Fig. 30.

Figure R2-28. The *in-situ* Raman spectra collected under WGS reaction condition (2%CO/3%H₂O/Ar) at 250 °C.

Figure R2-29. Apparent reaction order of H₂O and CO for the Co/CeO₂-PM catalysts.

REVIEWERS' COMMENTS

Reviewer #1 (Remarks to the Author):

This paper has been revised according to the referee's comments. This paper can be published in Nature Communications.

Reviewer #2 (Remarks to the Author):

All questions were addressed.